

# Driving mechanisms for the ENSO impact on stratospheric ozone

Samuel Benito-Barca[1], Natalia Calvo[1], Marta Abalos[1]

[1]Earth Physics and Astrophysics Department, Universidad Complutense de Madrid, Madrid, Spain

*Correspondence to*: Samuel Benito-Barca (samubeni@ucm.es)

5 **Abstract.** While the impact of El Niño-Southern Oscillation (ENSO) on the stratospheric circulation has been long recognized, its effects on stratospheric ozone have been less investigated. In particular, the impact on ozone of different ENSO flavors, Eastern Pacific (EP) El Niño and Central Pacific (CP) El Niño, as well as the driving mechanisms for the ozone variations have not been investigated to date. This study aims to explore these open questions by examining the anomalies in advective transport, mixing and chemistry associated with different El Niño flavors (EP and CP) and La Niña in the Northern Hemisphere in boreal winter. For this purpose, we use four 60-year ensemble members of the Whole Atmospheric Community Climate Model version 4. The results show a significant ENSO signal on total column ozone (TCO) during EP El Niño and La Niña events. During EP El Niño events, TCO is significantly reduced in the tropics and enhanced at middle and high latitudes in boreal winter. The opposite response has been found during La Niña. Interestingly, CP El Niño has no significant impact on extratropical TCO while its signal in the tropics is weaker than for EP El Niño events. The analysis of mechanisms reveals that advection through changes in tropical upwelling is the main driver for ozone variations in the lower tropical stratosphere, with a contribution of chemical processes above 30 hPa. At middle and high latitudes, stratospheric ozone variations related to ENSO result from combined changes in advection by residual circulation downwelling and changes in horizontal mixing linked to Rossby wave breaking and polar vortex anomalies. The impact of CP El Niño on the shallow branch of the residual circulation is small, and no significant impact is found on the deep branch.

## 1 Introduction

El Niño-Southern Oscillation (ENSO) is one of the main sources of interannual variability in the global climate. Although this phenomenon takes place in the Tropical Pacific Ocean, its impacts reach the stratosphere (e.g. García-Herrera et al., 2006; Manzini et al., 2006; Calvo et al., 2017; see Domeisen et al., (2019) for a review). During boreal winter, El Niño (the warm ENSO phase) signal can propagate poleward from the tropical Pacific by means of atmospheric Rossby wave trains. In the Northern Hemisphere (NH), this is related to a deeper Aleutian low and a strengthening of the Pacific-North American (PNA) pattern. As a consequence, the propagation of Rossby waves into the stratosphere is enhanced through the intensification of stationary wave number 1 (Manzini et al., 2006). Increased upward propagation of planetary waves during El Niño into the stratosphere results in a weakened polar vortex and a strengthening of the residual circulation of the Brewer-Dobson Circulation (BDC), which leads to tropical stratospheric cooling and stratospheric polar cap warming (e.g. Calvo et al., 2010;





Mezzina et al., 2021). In contrast, during La Niña, a weakening of the Aleutian low and destructive linear interference with the climatological wave pattern occur, resulting in a stronger and colder NH polar vortex and a weakening of the residual circulation (Iza et al., 2016).

Recently, the importance of distinguishing between two flavors of El Niño has arisen. For the canonical El Niño or Eastern Pacific El Niño, sea surface temperatures (SSTs) anomalies peak in the eastern equatorial Pacific, while El Niño Modoki,

Dateline El Niño or Central Pacific El Niño is characterized by SSTs anomalies that peak in the central equatorial Pacific (Larkin and Harrison, 2005; Ashok et al., 2007; Kao and Yu, 2009). In our study, we will use Eastern Pacific (EP) El Niño and Central Pacific (CP) El Niño to denote these two types of El Niño events.

Most studies have considered the impacts of EP El Niño as the canonical response to the warm phase of ENSO. However, fewer studies have examined the NH stratospheric response to CP El Niño and their results were many times contradictory.

On the one hand, Hegyi et al., (2014) and Hurwitz et al., (2014) found a similar response in the NH polar stratosphere to both CP and EP El Niño events (a weaker and warmer polar vortex) using idealized simulations with WACCM4 and studying the seasonal mean polar cap geopotential anomaly at 50 hPa in a set of CMIP5 models, respectively. Garfinkel et al., (2013) and Weinberger et al., (2019) also found that both EP and CP El Niño lead to a weakening of the polar vortex, but weaker during CP El Niño in early winter. On the other hand, Xie et al., (2012) -using reanalysis data- found the opposite signal (albeit of

smaller amplitude) for CP El Niño compared to EP El Niño, while Calvo et al., (2017) using CMIP5 models did not find a robust response to CP El Niño events in the extratropical NH stratosphere. The results shown in Calvo et al., (2017) highlighted the importance of studying the seasonal evolution of the NH stratospheric signals for understanding the different EP and CP El Niño impacts. Several reasons have been proposed to explain the contradictory results among studies. Garfinkel et al., (2013) concluded that the sign of NH stratospheric response to CP El Niño depends on the index used to identify CP El Niño

events, the composite size and the month average analyzed. Other reasons may include interactions between El Niño and the Quasi-Biennial Oscillation (QBO; Xie et al., 2012) and overlapping with the signal from Sudden Stratospheric Warmings (SSWs; Iza and Calvo, 2015). Overall, further investigation is needed to clarify the differences between EP and CP El Niño signals on the NH stratosphere.

Stratospheric ozone is an important component of the climate system and plays a key role in the radiative budget and protecting

the Earth from the harmful solar ultraviolet (UV) radiation. In recent years several studies have reported that polar stratospheric ozone changes and extremes can exert significant influence on the NH surface climate (Calvo et al., 2015; Ivy et al., 2017; Stone et al., 2019). Despite its importance, few studies have addressed the impact of ENSO on stratospheric ozone in depth. Most of them mainly focused on the anomalously low ozone values in the tropical lower stratosphere during the ENSO warm phase associated with anomalously strong tropical upwelling (Marsh and Garcia, 2007; Randel et al., 2009; Calvo et al., 2010;

Oman et al., 2013). However, the impact of ENSO on ozone is not restricted to the tropical stratosphere. Changes in the BDC due to anomalous Rossby wave dissipation during ENSO events are linked to ozone anomalies in NH mid-latitudes and the polar region opposite to those in the tropics (Cagnazzo et al., 2009; Diallo et al., 2019; Lin and Qian, 2019).





Despite the ENSO signal on stratospheric ozone is clear, there are still many open questions. First of all, the driving mechanisms for these ozone anomalies remain unknown. Previous studies assumed that changes in the residual circulation of the BDC drives the anomalous ozone concentrations during ENSO events. However, global distribution of ozone is driven not only by advection due to residual circulation, but also by isentropic mixing following Rossby wave dissipation, as well as by chemical production and loss (Garcia and Solomon, 1983; Plumb, 2002; Abalos et al., 2013). In fact, the importance of mixing on stratospheric tracer transport, and in particular on the distribution of ozone, has been increasingly recognized (Salby and Callaghan, 2007; Garny et al., 2014; Dietmüller et al., 2017). Hence, anomalous ENSO-related ozone concentrations are expected to be generated by a balance between changes in advection by the residual circulation, changes in mixing related to wave dissipation and also changes in chemistry through the ENSO modulation of stratospheric temperatures and concentration of other chemical species. A second open question is whether different ENSO flavors can affect ozone concentrations differently, and whether the driving mechanisms are the same or differ between EP and CP El Niño events. Indeed, it is expected that if different response appears during EP and CP El Niño events in stratospheric temperature, polar vortex or planetary wave activity, this has an impact on the ozone response, and in particular on advection, mixing and chemistry.

The present study constitutes the first comprehensive analysis of the NH stratospheric ozone signal and driving mechanisms in response to different El Niño flavors (EP and CP El Niño) and La Niña in boreal winter. The analysis of simulations from the Whole Atmosphere Community Climate Model (WACCM), a chemistry-climate model with a well resolved stratosphere, allows us to evaluate the separate contributions of the advective BDC, the isentropic mixing and the chemical processes to ozone variations during ENSO events. In the remaining of the paper, the methodology, model simulations, reanalyses and observational dataset analyzed are described in Sect. 2. Section 3 analyzes the seasonal mean impact of ENSO events on the NH stratosphere and the monthly evolution of the total column ozone (TCO). The driving mechanisms of the anomalous ozone concentration are examined in Sect. 4 while Sect. 5 summarizes the main conclusions of this study.

## 2    Data and Methods

We use monthly averaged fields from four ensemble members (60 years each, a total of 240 years) of the Whole Atmosphere Community Climate Model (WACCM4, Marsh et al., 2013; Garcia et al., 2017). This WACCM version has a horizontal resolution of 1.9º latitude by 2.5º longitude and 66 levels in the vertical with the top at about 140 km. These simulations, which were carried out for the Chemistry-Climate Model Initiative (CCMI, Eyring et al., 2013), were performed with prescribed observed SSTs and external forcings to match the observations for the period 1955-2014 (CCMI REFC-1 configuration). The QBO was nudged by relaxing the stratospheric tropical zonal winds towards observations.

In order to eliminate the influence of the QBO, we performed a multiple linear regression analysis to the simulated time series. Following Wallace et al. (1993), we use two QBO indices corresponding to the first two empirical orthogonal functions (EOFs) of the zonal wind between 5º S and 5º N over the layer 10-70 hPa. The results of the multiple regression fit are subtracted from the original data; then we use the residual series, which contain the ENSO signal, for our analysis.



ENSO events are identified directly from the observational record, since the WACCM simulations analyzed here have been run with observed SSTs. EP El Niño and CP El Niño events are selected as in Iza and Calvo, (2015). El Niño events are defined using the standardized November to February (NDJF) SSTs anomalies in the Niño3 (N3, 5º N-5º S, 150º W-90º W) and Niño4 (N4, 5º N-5º S, 160º E-150º W) regions. EP El Niño events are selected when N3 exceeds 0.5 standard deviations (std) and N3 minus N4 is larger than 0.1 std. Analogously, CP El Niño events are selected when N4 exceeds 0.5 std and N4 minus N3 is

larger than 0.1 std. For La Niña events, we follow the criteria of Iza et al., (2016) for "strong" La Niña events. Using the standardized NDJF SSTs anomalies in the Niño3.4 region (N3.4, 5º N-5º S, 170º W-120º W), strong La Niña events are identified when N3.4 is less than -1 std. We have used the N3.4 index for identified La Niña events since different La Niña flavors have not been established in the observational record.  We use the threshold of -1 std instead of -0.5 std to select La Niña winters since Iza et al., (2016) demonstrated that, using the threshold of -0.5 std, La Niña signal is masked by other

sources of variability like SSWs.  Table 1 lists the selected ENSO events used in this work.

**Table 1. Identified EP El Niño, CP El Niño and La Niña events. Numbers in brackets indicate the value of El Niño index used for selecting in each case (N3 for EP, N4 for CP and N3.4 for La Niña).**

| EP El Niño<br>N3 mean = 1.78 | CP El Niño<br>N4 mean = 1.16 | La Niña<br>N3.4 mean = -1.43 |
|---|---|---|
| 1965-1966 (1.11) | 1968-1969 (1.28) | 1970-1971 (-1.23) |
| 1972-1973 (1.76) | 1977-1978 (0.73) | 1973-1974 (-1.82) |
| 1976-1977 (0.87) | 1987-1988 (1.15) | 1975-1976 (-1.45) |
| 1982-1983 (2.73) | 1990-1991 (1.03) | 1988-1989 (-1.60) |
| 1986-1987 (1.14) | 1994-1995 (1.29) | 1998-1999 (-1.30) |
| 1997-1998 (3.10) | 2001-2002 (0.66) | 1999-2000 (-1.42) |
|  | 2002-2003 (1.24) | 2007-2008 (-1.40) |
|  | 2004-2005 (1.31) | 2010-2011 (-1.24) |
|  | 2006-2007 (1.23) |  |
|  | 2009-2010 (1.66) |  |


The ENSO signal is analyzed by compositing monthly mean anomalies for the identified ENSO events (Table 1) in boreal extended winter (October to March). Anomalies are computed with respect to a 21-year running mean climatology, which allows to remove possible linear and non-linear trends. This is particularly important in the case of ozone since Ozone Depleting Substances (ODSs) concentrations are not uniform throughout the 1955-2014 period. The statistical significance of

the ENSO signal in the composites is assessed with a Monte Carlo test of 1000 trials at the 95 % confidence level.



For model validation and comparison purposes, ozone data from two reanalyses and an observational dataset have been used. The same methodology applied to WACCM has been followed here to remove the influence of the QBO and obtain the ENSO signal. The Modern-Era Retrospective analysis for Research and Applications, Version 2 (MERRA-2, Gelaro et al., 2017) provides data at horizontal resolution of 0.5º latitude by 0.625º longitude and 42 pressure levels with the top at 0.1 hPa. For

our study, monthly mean ozone data on pressure levels covering the period January 1980-December 2016 have been used. MERRA2 calculates ozone concentrations as a fully prognostic variable, subject to assimilation, a photochemistry scheme and transport. It assimilates ozone satellite observations from NOAA's SBUV (Solar Backscatter Ultraviolet Radiometer) until 2004 and NASA's Aura OMI (Ozone Monitoring Instrument) and Aura MLS (Microwave Limb Sounder) afterwards. MERRA2 ozone concentrations generally show better agreement with observations than other reanalyses, especially in the

middle stratosphere (Davis et al., 2017).

The Japanese 55-year reanalysis (JRA55, Kobayashi et al., 2015) has also been used. JRA55 has a horizontal resolution of 2.5º latitude by 2.5º longitude and 37 pressure levels with the top at 1 hPa. In JRA55, ozone observations are not assimilated directly. Before 1979, a monthly mean climatology for the 1980-1984 period is used. From 1979 onwards, ozone fields are produced using an offline chemistry climate model (MRI-CCM1) that assimilates TCO observations from NASA's TOMS

(Total Ozone Mapping Spectrometer) until 2004 and Aura OMI afterwards using a nudging scheme (Shibata et al., 2005). In this study, we use JRA55 ozone for the period January 1980-December 2016.

The Stratospheric Water and Ozone Satellite Homogenized (SWOOSH) database is a merged zonal-mean monthly-mean dataset which contains observations from SAGE II (v7.0), SAGE III (v4), HALOE (v19), UARS MLS (v5) and EOS Aura MLS (v4.2) instruments (Davis et al., 2016). The SWOOSH dataset used in this study is version 2.6, with horizontal resolution

of 2.5º latitude and 31 vertical levels between 1 and 316 hPa covering the period January 1984-December 2016. We use specifically the "combinedanomfillo3q" product.

## 3    Stratospheric impact of ENSO

Before investigating the ozone behavior, we evaluate the ENSO response in temperature, zonal wind and residual circulation

in WACCM4 against results from previous literature. Figure 1a-c shows the latitude-pressure November-February (NDJF) anomalies of the zonal-mean temperature and zonal-mean zonal-wind composited for EP El Niño, CP El Niño, and La Niña events (Figs. 1a, b and c, respectively). In the tropics, both EP El Niño and CP El Niño signals are characterized by a significant warming in the troposphere and a cooling in the stratosphere, peaking at about -1.4 K between 50 and 70 hPa in EP El Niño and at about -1.2 K in CP El Niño. Along with these anomalies, a robust strengthening of the subtropical jets appears in both

EP El Niño and CP El Niño events, stronger during EP El Niño (at about 3 m s$^{-1}$ versus 2 m s$^{-1}$ in the NH). La Niña signal (Fig. 1c) is opposite to that of El Niño, characterized by a significant cooling in the troposphere, warming in the stratosphere and a weakening of the subtropical jets.

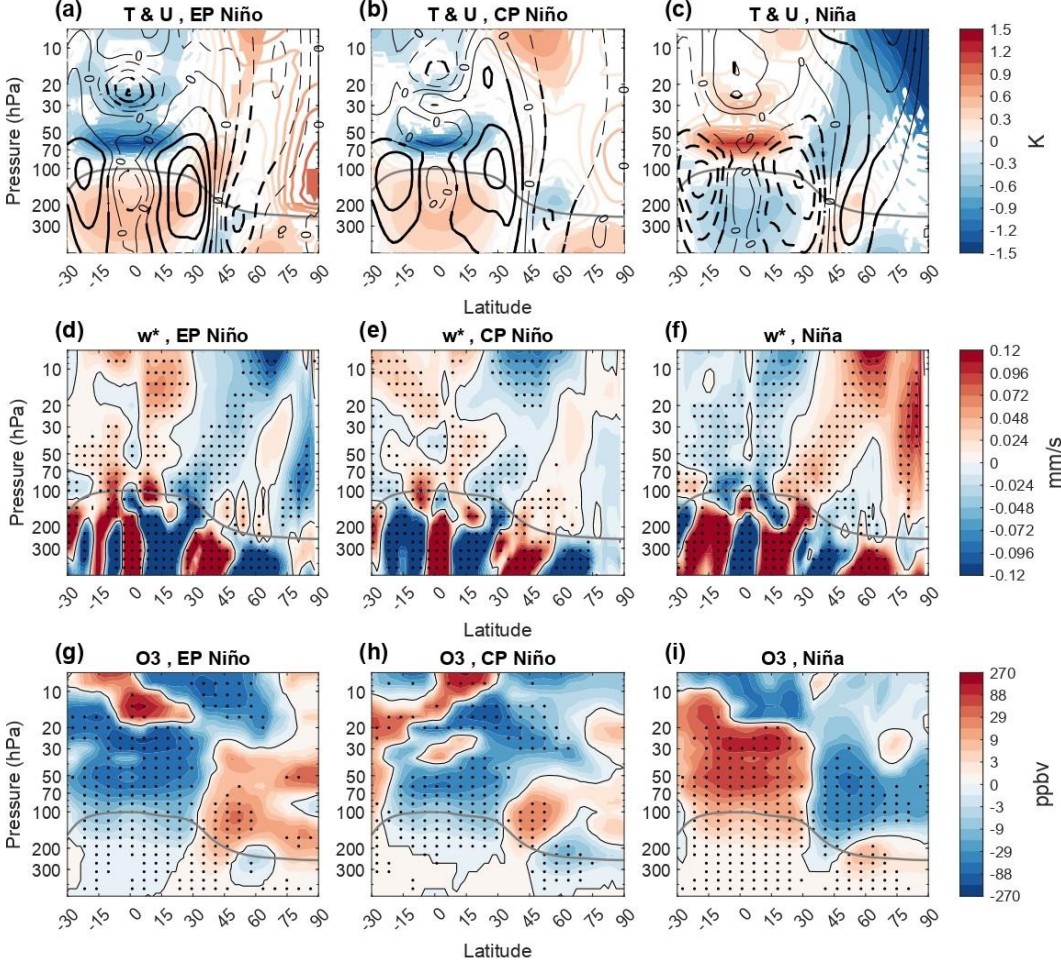

**Figure 1. Latitude - pressure cross sections of the composite of NDJF zonal mean (a-c) temperature (colors) and zonal wind (black contours), (d-f) vertical component of the residual circulation w\* and (g-i) ozone mixing ratio anomalies for (from left to right) EP El Niño, CP El Niño and La Niña events. Contours in upper panels are drawn every 0.6 m s-1 for zonal wind and 0.15 K for temperature. Solid (dashed) contours denote positive (negative) anomalies. The NDJF mean tropopause is indicated by the thick grey line. Color shading (upper panels) denotes statistically significant anomalies at the 95 % confidence level for temperature and black dots (middle and bottom panels) the same for w\* and ozone mixing ratio. Thick black contours (upper panels) denote statistically significant anomalies for zonal wind.**

At middle latitudes (~ 30°-60° N), EP El Niño and CP El Niño signals in the lower stratosphere show larger differences than in the tropics. Significant anomalies are only found during EP El Niño as anomalous warming. During La Niña events, the anomalies are opposite to those in EP El Niño, with a significant cooling in the lower and middle stratosphere. At high latitudes, the EP El Niño temperature response is characterized by warm anomalies in the polar stratosphere, only significant in the lowermost stratosphere, and a significant weakening of the polar vortex that extends into the troposphere. In contrast, the



temperature signal of CP El Niño events is not significant in the polar stratosphere, with anomalies in the polar vortex weaker than in the EP El Niño events. During La Niña events, a robust cooling appears in the middle and upper polar stratosphere accompanied by a strengthening of the polar vortex. The different location of the significant zonal-mean polar temperature anomaly between EP El Niño and La Niña is likely due to the occurrence of the SSWs. When the temperature response is analyzed only for winters without SSWs, the stratospheric warming associated with EP El Niño events also extends into the middle and upper stratosphere (not shown). Overall, the ENSO response shown here is in good agreement with previous knowledge from radiosonde studies (Free and Seidel, 2009), reanalyses data (García-Herrera et al., 2006; Camp and Tung, 2007; Iza and Calvo, 2015; Iza et al., 2016) and model simulations (Randel et al., 2009; Calvo et al., 2010, 2017, Diallo et al., 2019).

In addition to changes in temperature and zonal wind, ENSO has also impact on the residual circulation (Fig. 1d-f). During EP El Niño, a significant strengthening of the shallow and deep branches of the residual circulation occurs (Fig. 1d) in agreement with results from previous studies which analyzed the canonical response to ENSO (e.g. García-Herrera et al., 2006; Calvo et al., 2010). This is consistent with the ENSO signal in temperature as anomalously cold regions coincide with positive $w^*$ anomalies and viceversa. In contrast, during CP El Niño, only a slight acceleration occurs in the shallow branch, and no significant changes are simulated in the deep branch (Fig. 1e), consistent with the lack of significant CP El Niño signal in polar stratospheric temperature shown above. Regarding La Niña, $w^*$ anomalies pattern mirror that during EP El Niño, with a deceleration of the residual circulation (Fig. 1f) leading the tropical warming and the extratropical cooling. These results highlight differences between EP and CP El Niño events on the residual circulation and reveal that CP El Niño has no impact on the deep branch.

Next, we examine the ENSO anomalies on ozone, shown in Fig. 1g-i. They show robust changes in stratospheric ozone mixing ratio in response to ENSO. Both EP El Niño (Fig. 1g) and CP El Niño (Fig. 1h) events show a significant reduction of ozone mixing ratios in the tropics in the lower and middle stratosphere and an increase at middle latitudes only in the lower stratosphere, always stronger during EP El Niño events. At high latitudes, a significant increase in ozone concentrations appears in the lower stratosphere only during EP El Niño, in agreement with the lack of CP El Niño signal in zonal mean temperature shown above. The anomalous ozone pattern during La Niña events (Fig. 1i) is very similar to that of the EP El Niño but with opposite sign. All these results for the lower stratosphere are in line with previous studies using model simulations (Randel et al., 2009; Calvo et al., 2010), observations (Lin and Qian, 2019) and reanalyses (Diallo et al., 2019). However, none of them distinguished between EP and CP El Niño, while Fig. 1g and Fig. 1h clearly demonstrate that the anomalies are overall larger for EP El Niño than CP El Niño and in particular for the polar region CP El Niño events do not have a statistically significant effect. Thus, our results highlight the need to distinguish between the two types of El Niño to explore the impact of ENSO on the stratospheric composition, and specifically on stratospheric ozone concentrations.

Variations in ozone mixing ratio can potentially affect TCO and therefore the net UV levels reaching the surface. However, since anomalies in the ozone mixing ratios do not extend over the entire stratosphere, and anomalies of opposite sign appear at different stratospheric levels, it is unclear from Fig. 1 whether ENSO actually affects TCO throughout the winter. For this



purpose, Figure 2 shows the latitude-time October to March evolution of WACCM4 TCO anomalies composited for ENSO events. For comparison, two reanalyses (JRA55 and MERRA2, Fig. 2d-f and 2g-i, respectively) and satellite observations (SWOOSH, Fig. 2j-l) are included. Note that very few events are included in SWOOSH and reanalyses for EP El Niño, and

therefore the comparison in this case should be made with caution.

In the tropical region (~30° S-30° N), WACCM shows a significant reduction of TCO during EP El Niño events and, to a lesser extent, during CP El Niño events, while an increase of TCO appears during La Niña events. The comparison of WACCM simulations with reanalysis and observations reveals particularly good agreement for La Niña events, as the positive anomalies are significant in both reanalyses and SWOOSH despite the small composite size (five cases). For EP El Niño events significant

negative anomalies in the tropics are found from December to March in reanalyses, while SWOOSH does not show any significant anomalies. Note that unfortunately the composite with SWOOSH data includes only two EP El Niño events, so the significance in this case needs to be taken with caution. Results for CP El Niño events are less robust. While model, reanalyses and observations show negative anomalies in the tropics, weaker than those for their corresponding EP El Niño composites, the seasonality and statistical significance differs across datasets. In particular, only WACCM and MERRA2 show significant

anomalies.

At mid-latitudes (~30-60° N), WACCM shows an increase in TCO during EP EL Niño events from December to March, the opposite during La Niña events. These results agree well with reanalyses and observations in February and March. Interestingly, CP El Niño events show no significant signal in this region in any of the datasets. As shown above, the positive anomalies in ozone mixing ratio (Fig. 1h) that appear in CP El Niño events in the lower mid-latitude stratosphere are weaker

than in EP El Niño and are also accompanied by anomalies of opposite sign in the mid-stratosphere. This dipole structure at mid latitudes leads to a lack of significant signal in TCO for CP El Niño at middle latitudes.

At high latitudes (~ 60-90° N), WACCM shows significant positive TCO anomalies in late winter during EP El Niño events, while during La Niña events the TCO response is opposite to that. This is in good agreement with SWOOSH and reanalyses, although the anomalies do not reach significance in these datasets likely due to the few cases composited and the large

variability of the polar stratosphere. Differences in early winter in EP El Niño between WACCM and reanalyses could also come from variability of the polar stratosphere related to SSWs. The larger occurrence of SSWs in November in WACCM during EP El Niño favors positive anomalies to appear earlier in the model than in reanalyses. Results for CP El Niño events are more uncertain in this region. WACCM shows a reduction in TCO in late winter, which does not appear in reanalyses or SWOOSH.




**Figure 2. October to March composite evolution of total column ozone (TCO) anomalies (DU) as a function of latitude in (a-c) WACCM, (d-f) JRA55, (g-i) MERRA2 and (j-l) SWOOSH for (from left to right) EP El Niño, CP El Niño and La Niña events. Numbers in brackets indicate the number of events in each composite. Black dots denote statistically significant anomalies at the 95 % confidence level.**





Therefore, the high-latitude TCO signal during CP El Niño events seems to be weaker and more uncertain and no general conclusions can be drawn in this region, consistent with the lack of CP El Niño signal in the seasonal mean zonal mean temperature, deep branch of the residual circulation and ozone mixing ratio.

In summary, the analysis above shows that both EP El Niño and La Niña have a robust signal on TCO. These results are in line with Cagnazzo et al., (2009), who found an increase in the polar TCO and a reduction of tropical TCO associated in satellite observations and a set of Chemistry Climate Models but only for the canonical El Niño. Here we have shown that La Niña also has an impact on TCO and also that the signal of CP El Niño appears only in the tropics but not at high latitudes. Larger anomalies observed in reanalyses than in WACCM are likely due to the lower number of events composited in the

former. In fact, when individual events are considered, the magnitude of the anomalies is similar in WACCM, reanalyses and SWOOSH (not shown). Therefore, the overall good agreement between WACCM and reanalyses data allows us to use WACCM simulations to investigate the mechanisms that are controlling the ozone changes during ENSO events in the next section.

## 4    Driving mechanisms of ozone during ENSO

As discussed in the Introduction, the robust ENSO changes in stratospheric ozone shown in the previous section can be caused by advection due to residual circulation, isentropic mixing following planetary wave dissipation and/or local chemical production and loss. We use WACCM simulations to evaluate the different terms of the TEM continuity equation for zonal-mean ozone concentration (Eq. 1). This equation provides the local change in ozone concentration as a result of transport and chemical processes (Andrews et al., 1987).

$$\overline{\chi_t} = -\overline{v^*\chi_y} - \overline{w^*\chi_z} + e^{\frac{z}{H}}\nabla \cdot M + P - L \qquad (1)$$

In Eq. (1), overbars denote zonal means and subindices indicate partial derivatives. The term on the left-hand side represents the local tendency in ozone concentration, where $\chi$ indicates the ozone mixing ratio. On the right-hand side, the first and second terms represent the advection due to residual circulation $(v^*, w^*)$, P-L is the ozone tendency due to chemistry (chemical production minus loss rate) and $e^{\frac{z}{H}}\nabla \cdot M$ denotes the eddy transport term, whose horizontal and dominant component is related

to isentropic mixing, represented as the divergence of the eddy transport vector $\boldsymbol{M}=(0, M_y, M_z)$, with components defined as in Andrews et al., (1987):

$$M_y = -e^{\frac{-z}{H}}\left(\overline{v'\chi'} - \frac{\overline{v'T'}}{S}\overline{\chi_z}\right) \qquad (2)$$

$$M_z = -e^{\frac{-z}{H}}\left(\overline{w'\chi'} - \frac{\overline{v'T'}}{S}\overline{\chi_y}\right)$$

where primes indicate deviations from zonal means, T is the air temperature and S = $N^2$ * H / R with H = 7 km, R = 287 $m^2 s^{-2} K^{-1}$ and $N^2$ the Brunt-Väisälä frequency.






The analysis of the different terms of Eq. (1) has been carried out as follows: first, ozone concentration anomalies are examined considering 3 different regions: tropics (20º S-20º N), mid-latitudes of the NH (35º-55º N) and Arctic region (70º-90º N). Second, the anomalies in the local tendency of ozone concentration (left term in Eq. 1) are obtained and finally the terms on the right-hand side of Eq. (1) are analyzed to understand the driving mechanisms that give rise to the ozone anomalies. This

analysis has been carried out using three of the four members of the WACCM4 ensemble since data for the eddy transport term was not available in the fourth.

Figure 3 displays the time-pressure evolution of ozone mixing ratio anomalies averaged over each of the three regions defined above for the three ENSO types. First, we analyze the anomalies in the tropics (Fig. 3a-c).

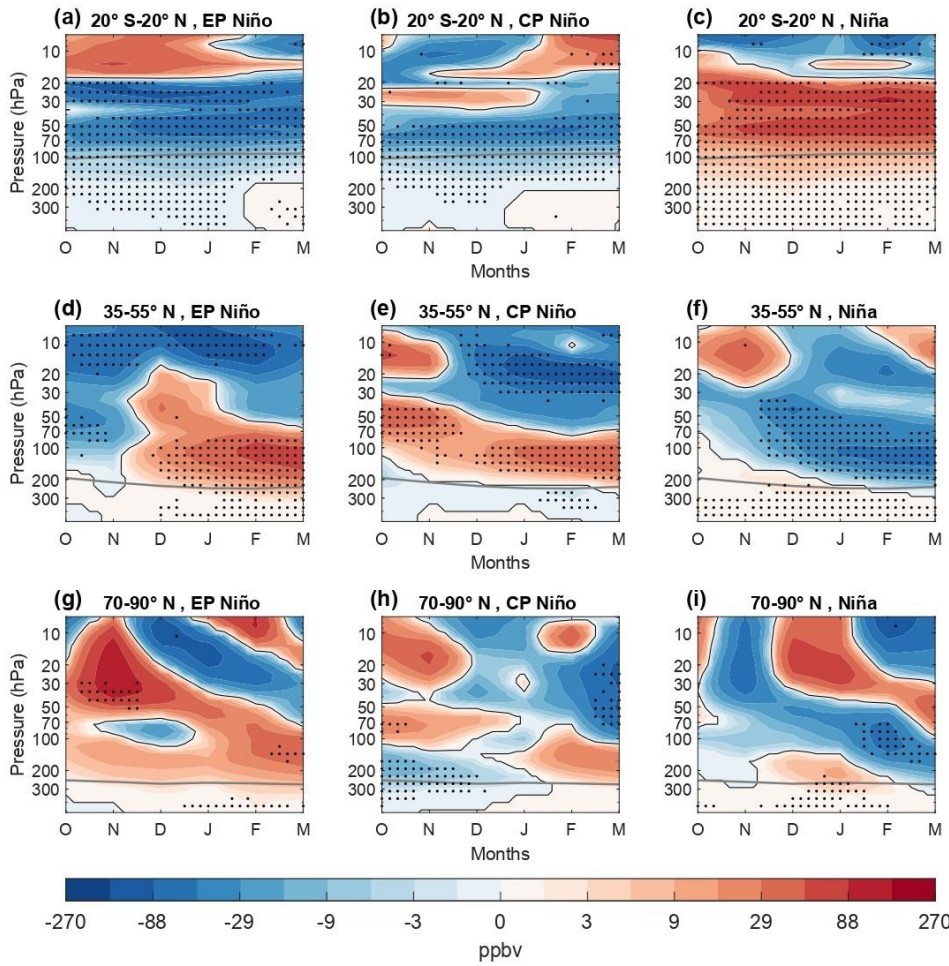

**Figure 3. October to March composite evolution of ozone mixing ratio anomalies (ppbv) as a function of pressure (a-c) in the tropics (20º S-20º N), (d-f) at mid-latitudes (35º-55º N) and (g-i) in the Arctic (70º-90º N) for (from left to right) EP El Niño, CP El Niño and La Niña events. Black dots denote statistically significant anomalies at the 95 % confidence level. The tropopause is indicated by the thick grey line.**



As expected from Fig. 1 and Fig. 2, robust negative ozone anomalies during EP El Niño and positive anomalies during La Niña events are present throughout the entire winter in the lower and middle tropical stratosphere (below 20 hPa). In CP El Niño events, anomalies are weaker than in EP El Niño and confined below 50 hPa.

In order to understand the driving mechanisms of these tropical anomalies, Figure 4 shows the anomalies in the relevant terms of Eq. (1) for the tropical region. The anomalies in the ozone tendency are small (Fig. 4a-c), consistent with the near-constant

tropical ozone concentration anomalies throughout the winter in all three ENSO cases, as seen in Fig. 3a-c. It is clear that the anomalies in the tropical ozone tendency during ENSO events below 30 hPa come mainly from advection (Fig. 4d-f). This is consistent with anomalous tropical upwelling present in Fig 1d-f. Previous studies already showed increased tropical upwelling associated with El Niño events (e.g. Calvo et al., 2010; Diallo et al., 2019). Enhanced upwelling during El Niño leads to ozone-poor air rises from the tropopause region, where the ozone concentration is more than an order of magnitude lower, into the

stratosphere, generating negative anomalies therein. During La Niña events the response is the opposite: there is a decrease in tropical upwelling (Calvo et al., 2010) and less ozone-poor air reaches the stratosphere, leading to positive anomalies in ozone mixing ratio.

Above 30 hPa, ozone changes due to advection are counteracted by changes due to chemical processes (Fig. 4g-i). Hood et al., (2010) indicated that enhanced tropical upwelling following El Niño events leads to a reduction in odd nitrogen (NOx) in the

middle stratosphere. Such NOx decrease may lead to photochemical ozone increases by modifying the NOx ozone loss catalytic cycle. Co-occurrence of anomalies in the tendency due to advection and due to chemistry supports this hypothesis. This mechanism may also be acting during La Niña events. The reduction of tropical upwelling leads to a higher concentration of NOx in the middle stratosphere, and thus to a higher catalytic destruction of ozone. The eddy transport term tends to counteract the advection term below 30 hPa, consistent with the gradient-eroding effect of mixing, but the magnitude is smaller

(not shown o complementary). Regarding comparison between EP and CP El Niño events, the different strength and timing in the advection and chemistry anomalies are due to the differences in the intensification of tropical upwelling shown in Fig 1d-e and in the timing of occurrence of this enhanced upwelling. The largest anomalies in the tropical upwelling during EP El Niño occur in early winter, but during CP El Niño events the response mainly occurs after December (Not shown).

We next examine ozone anomalies at mid-latitudes (35-55º N, Fig. 3d-f). We focus especially on the anomalies located below

30 hPa, since these are the ones that have the largest impact on TCO. From December onwards, significant positive ozone concentration anomalies appear in the lower stratosphere associated with EP El Niño events, and negative anomalies associated with La Niña events. These ozone mixing ratio anomalies produce the TCO anomalies seen in Fig. 2a, c. During CP El Niño events, significant positive anomalies also appear in the lower stratosphere, although weaker than during EP El Niño events. Moreover, they are accompanied by strong negative anomalies above, between 15 and 30 hPa. This results in a lack of signal

in TCO at mid-latitudes during CP El Niño events as shown in Fig.2b.

The evaluation of the anomalous patterns of the terms in the right-hand side of Eq. (1) reveals that at mid-latitudes both advection due to the shallow branch of the residual circulation (Fig. 5d-f) and mixing (Fig. 5g-i) are key in generating the anomalies below 30 hPa, with both mechanisms leading to ozone changes (Fig. 5 a-c) of the same sign. During EP El Niño,

ozone accumulation occurs mainly in November and December (Fig. 5a) due to contribution of mixing in both months and

contribution of advection in December. During La Niña, negative ozone anomalies are generated from November to February

(Fig. 5c). In this case, the onset of the anomalies is dominated by advection, while mixing contributes from January

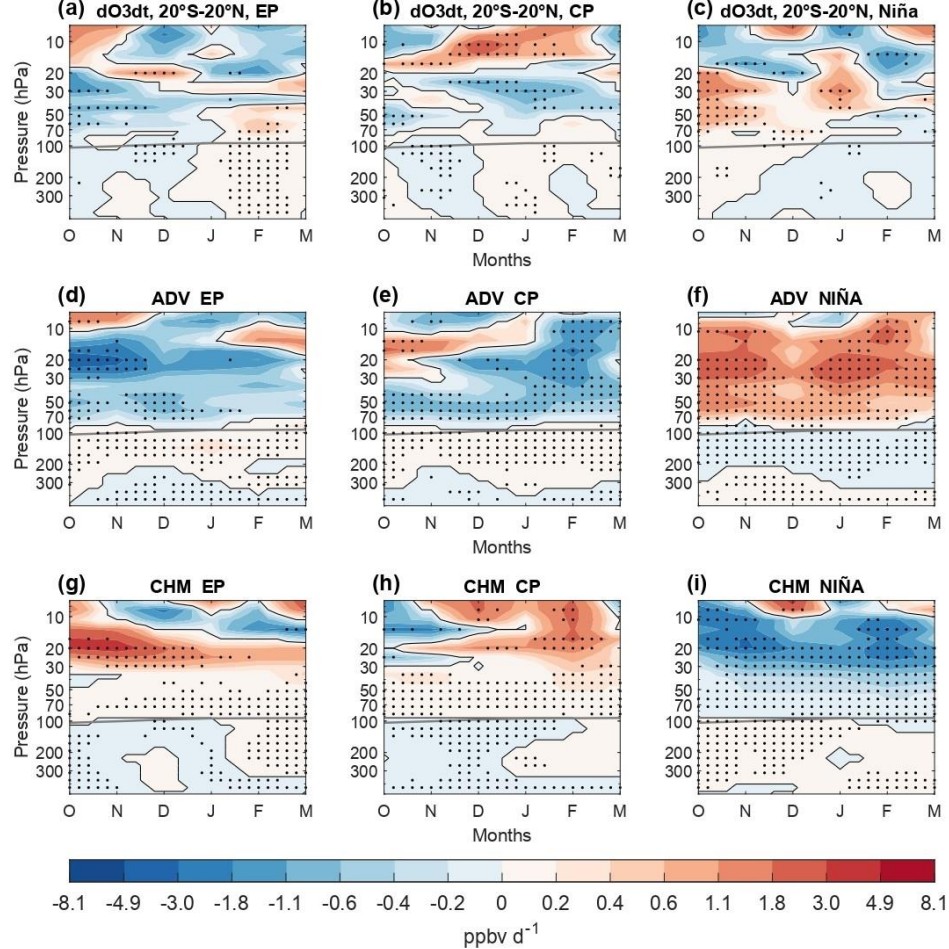

**Figure 4. October to March composite evolution of the anomalies of the most relevant terms in the zonal-mean ozone continuity equation (Eq. 1) as a function of pressure, average over 20ºS-20ºN, for (from left to right) EP El Niño, CP El Niño and La Niña**

**events. DO3dt (a-c), is the local tendency in ozone mixing ratio, ADV (d-f) is variation due to the advection and CHM (g-i) denote the chemical balance. Black dots denote statistically significant anomalies at the 95 % confidence level. The tropopause is indicated by the thick grey line.**

onwards. In CP El Niño events, weaker positive ozone tendency anomalies appear in January, mainly due to changes in mixing.

The negligible role of advection in the lower stratosphere during CP El Niño (Fig. 5e) contrasts with its larger role during EP

El Niño and La Niña.  This key result is consistent with the weak acceleration of the shallow branch during CP El Niño winters



than during EP El Niño in WACCM discussed in Sect. 3 (Fig. 1d-e). Chemical changes do not contribute significantly to the ENSO-related ozone anomalies in the midlatitudes below 30 hPa (not shown).

Having established the key role of mixing processes as a main driver of stratospheric ozone changes during ENSO events at
middle latitudes, we next study the spatial pattern of these anomalies and the factors that favor their occurrence. For this purpose, Figure 6 shows the latitude-pressure anomalies of the third term of Eq. (1), associated with mixing (Fig.6 a-c), and the Eliassen-Palm flux divergence (hereafter EPFD) and zonal mean zonal wind (Fig.6 d-f). Based on the timing of the largest mixing contribution below 30 hPa at middle latitudes (Fig.5 g-i), composites of EP El Niño anomalies are computed for the NDJFM average while CP El Niño and La Niña composites are computed for the JFM mean. The Eliassen-Palm flux is a
measure of planetary wave propagation, while EPFD is a measure of its dissipation (Andrews et al., 1987), with negative values of the EPFD indicating wave breaking. Planetary wave breaking is closely related to isentropic mixing, as it leads to

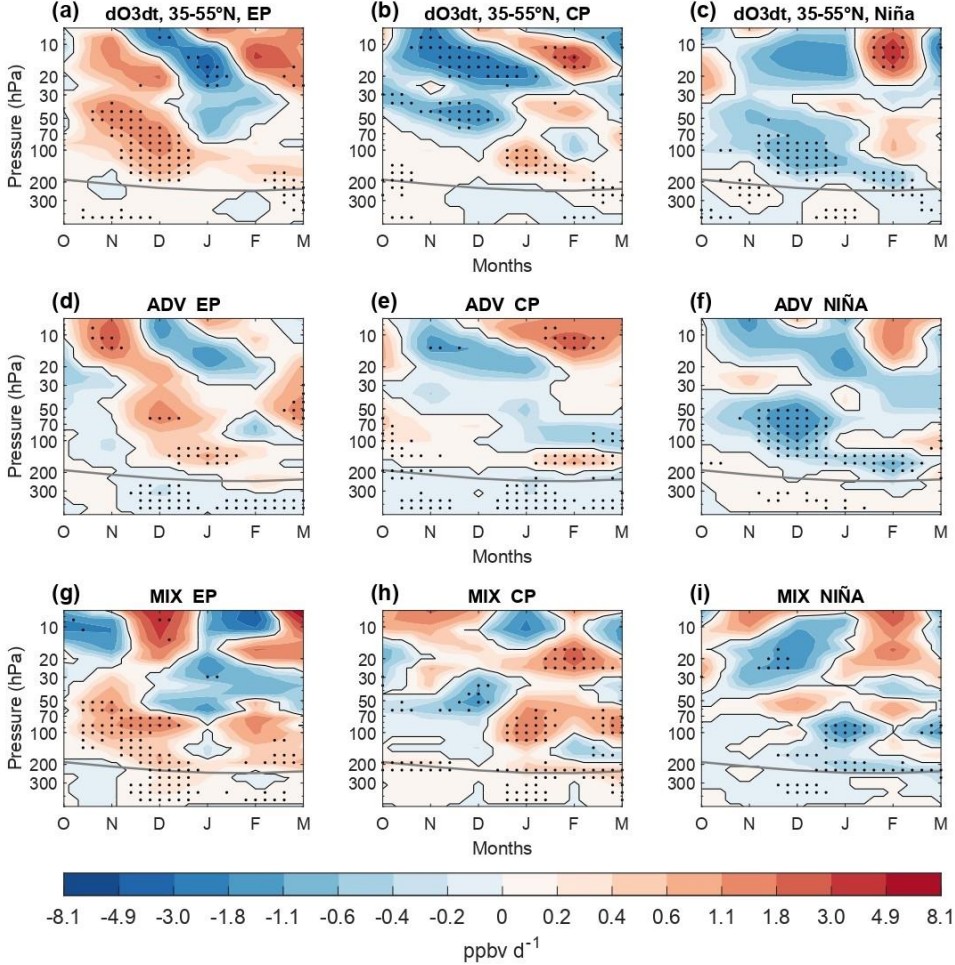

**Figure 5. As in Fig. 4 but average over 35-55º N and for anomalies in DO3dt (a-c), ADV (d-f), and MIX (g-i), which represents changes related to mixing.**





the development of tracer filaments which are ultimately diffused and mixed with the environment. The intensity of the polar vortex, directly linked to wave dissipation, constitutes a mixing barrier (Plumb, 2007), such that enhanced wave dissipation and mixing are related to a weak polar vortex and viceversa.

In all three ENSO cases (EP and CP El Niño, and La Niña), an anomalous dipole structure appears in the mixing term between mid-latitudes and polar latitudes below 50 hPa (Fig. 6a-c), suggesting that changes in one region are related to changes in the

other. To understand these variations, note that climatological ozone values below 30 hPa are higher at the pole than at mid-latitudes in boreal winter (not shown). Therefore, the climatological mixing effect tends to reduce this ozone concentration gradient generating a net transport of ozone from polar latitudes to mid-latitudes. Both EP and CP El Niño composites show positive anomalies in the mixing term at middle latitudes and negative anomalies in the polar region, indicating an intensification of quasi-horizontal mixing. Hence, the net effect of mixing during both types of El Niño events is to transport

more ozone from polar latitudes to mid-latitudes. These changes in mixing are driven by anomalous Rossby wave breaking as shown by negative values of the EPFD anomalies in the stratosphere in the region centered around 50-60° N, accompanied by a weaker polar vortex (Fig. 6d, e). In contrast, during La Niña events, anomalies in the mixing term indicate accumulation of ozone at the pole and reduction of ozone at mid-latitudes, therefore implying a net reduction of mixing. Likewise, during La Niña events, stratospheric wave breaking is reduced, resulting in a stronger polar vortex (Fig. 6f).

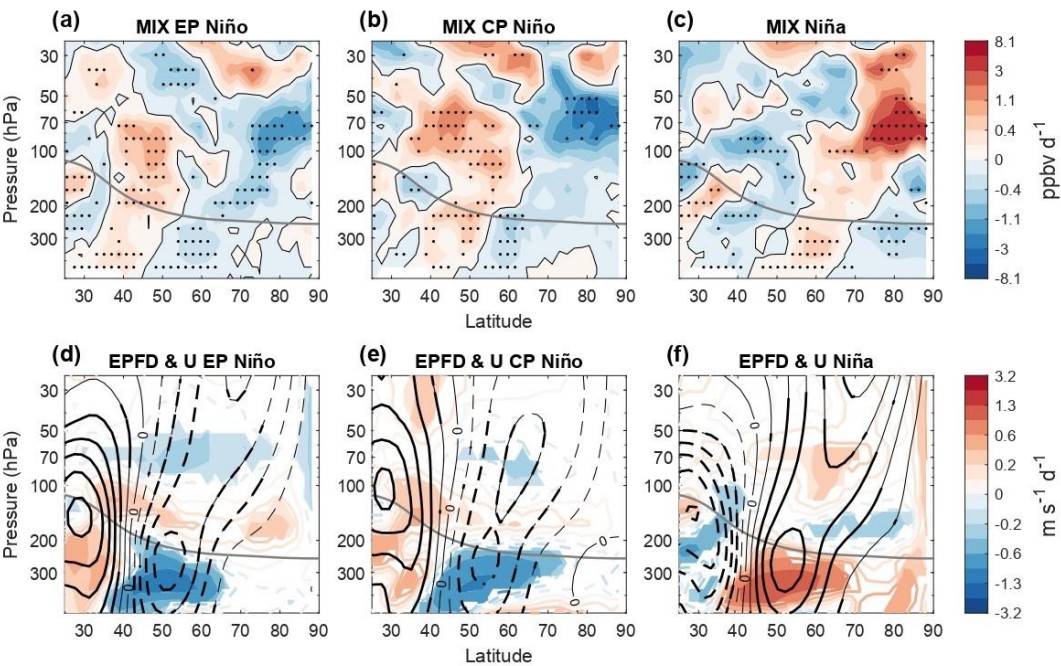


**Figure 6. Latitude-pressure cross sections of the composite of NDJFM zonal mean (EP El Niño) and JFM (CP El Niño and La Niña) (a-c) Ozone tendency related to mixing and (d-f) EPFD (colors) and zonal wind (black contours) for (from left to right) EP El Niño, CP El Niño and La Niña events. Contours in bottom panels are drawn every 0.5 m s-1 for zonal winds. Solid (dashed) contours denote positive (negative) anomalies. The NDJFM or JFM mean tropopause is indicated by the thick grey line. Color shading (bottom**
**panels) and black dots (upper panels) denote statistically significant anomalies at the 95 % confidence level; thick contours denote statistically significant anomalies for zonal wind.**



In summary, it is clear that the enhanced wave breaking around the polar vortex during EP El Niño and CP El Niño events causes an increase in mixing through a weakened polar vortex. Opposite changes occur during La Niña. The importance of the

wave-mean flow interaction on ENSO signals has been reported before using model simulations (e.g. Calvo et al., 2008; Li and Lau, 2013) and reanalyses data (e.g. Iza et al., 2016), but until now it had not been directly linked to mixing during ENSO events. Furthermore, our analysis demonstrates the importance of considering mixing as a key factor in ozone variations in the mid-latitude lower stratosphere during ENSO events, since its contribution to these changes is comparable to the advection by the shallow branch of the residual circulation, even more important during CP El Niño.

Finally, the dynamical mechanisms that control the changes in stratospheric ozone during different ENSO phases are analyzed in the Arctic region (70-90º N). Significant positive ozone anomalies appear in the middle stratosphere in early winter during EP El Niño (Fig. 3g) and propagate downward during winter to the lower stratosphere. Anomalies during La Niña events (Fig. 3i) are opposite to those during EP El Niño. These anomalies are consistent with the ones in TCO. During early winter, ozone mixing ratio anomalies in the lower stratosphere are weak, and therefore their impact on the TCO is small. Ozone mixing ratio

anomalies in the middle stratosphere have a minor impact on the TCO, and hence the TCO anomalies are generally small and not significant. In late winter, ozone concentration anomalies are significant in the lower stratosphere, but are partially offset by anomalies of opposite sign just above, weakening the impact on polar TCO. The anomalies during CP El Niño (Fig. 3h) are statistically insignificant in general, and no conclusions are drawn for this case.

Figure 7 shows the different terms involved in the high latitude ozone anomalies. The main driver of the downward propagating

anomalies in EP El Niño and La Niña is advection by the deep branch of the residual circulation (Fig.7 d-f). During EP El Niño events, the enhanced advection (Fig. 7d) accumulates ozone in the lower polar stratosphere as a result of the acceleration of the deep branch (Fig. 1d). However, in CP El Niño events, the effect of the advection is smaller and not significant (Fig. 7e), in agreement with the lack of significant CP El Niño impact on the deep branch of the residual circulation (Fig. 1e). During La Niña events the signal has opposite sign, with a deceleration of the deep branch and therefore, weaker ozone advection to

the polar lower stratospheric. Note that, contrary to middle latitudes, the effect of mixing is the opposite to that of advection for all ENSO composites (Fig.7 g-i). As shown in Fig. 6, negative anomalies in the mixing term at polar latitudes, as seen for EP and CP El Niño, imply that mixing between middle and polar latitudes increases, with more ozone being transported to mid-latitudes. In early winter, contributions of advection and mixing are balanced but in January and February, when anomalies in the deep branch of the residual circulation increase, advection is the dominant mechanism in the generation of anomalies.

Chemical net production is not an important factor in ozone anomalies at the pole since its action inside the polar vortex starts in spring (from March onwards) under the presence of solar radiation (not shown).

In summary, the analysis of the driving mechanisms of ozone variations during ENSO events has revealed advection as the main driver due to changes in the residual circulation. However, changes in advection alone cannot explain the ozone anomalies, and changes in chemistry (in the tropics above ~ 30 hPa) and mixing (at middle and high latitudes) must be





considered. Moreover, during CP El Niño advection in the extratropics is weaker, and mixing is the dominant transport term
in these regions.

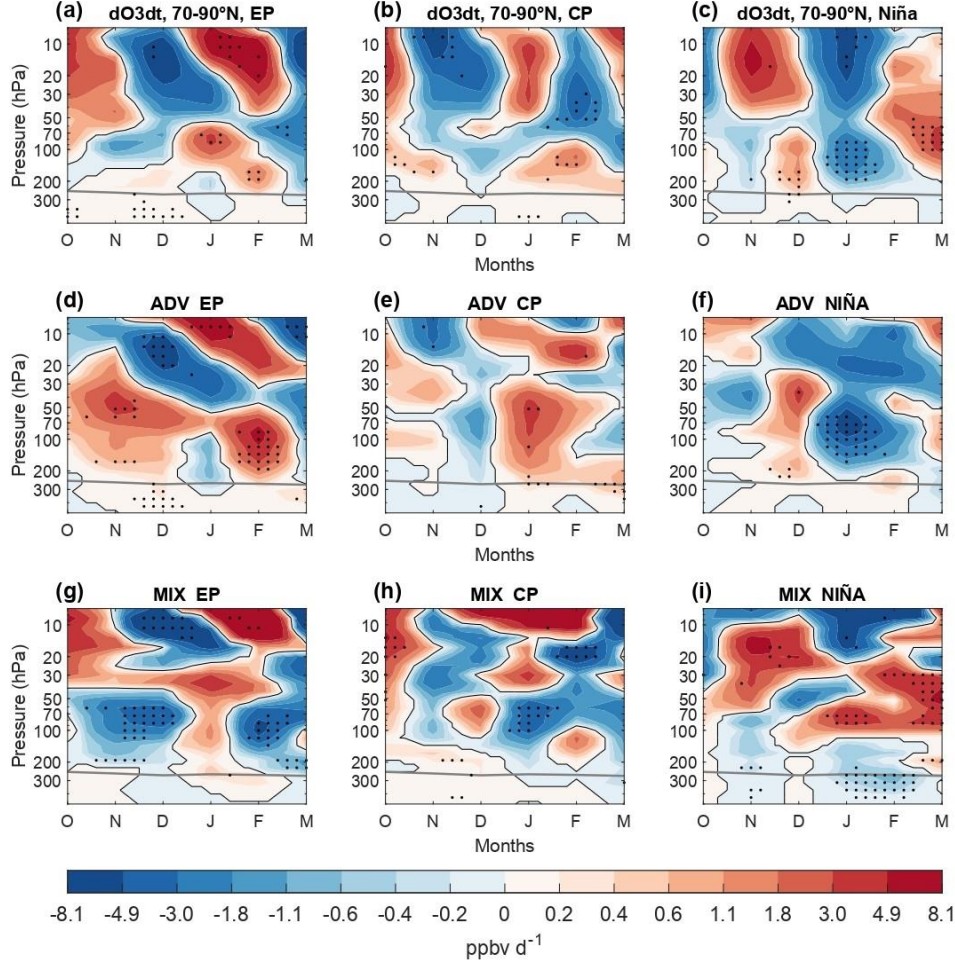

**Figure 7. As in Fig. 5 but average over 70-90º N.**

## 5   Summary and conclusions

In this study we analyzed NH ozone changes associated with ENSO phenomena in boreal winter, distinguishing for the first
time between different El Niño flavors (EP and CP El Niño) and La Niña. We used WACCM4 simulations with prescribed
observed SSTs and external forcings for the period 1955–2014 and analyzed four ensemble members to increase statistical
significance of the results. We evaluated the different terms in the continuity equation for zonal-mean ozone concentrations to
examine the driving mechanisms of ozone variations, separating contributions from the advective BDC, isentropic mixing and

chemical processes. Our results with WACCM confirm the importance of separately studying EP and CP El Niño events and



highlight the key role of mixing for middle and high latitude ozone variations during ENSO events. The main findings are summarized here:

- Both EP and CP El Niño events show, in the tropics, a robust impact on boreal winter temperature, zonal wind and ozone mixing ratio of the same sign but anomalies are larger for EP El Niño events. In contrast, only EP El Niño events show a significant impact in the Arctic region. In addition, both shallow and deep branches of the residual circulation are accelerated during EP El Niño (and decelerated during La Niña). However, during CP El Niño the shallow branch acceleration is up to three times smaller than in EP El Niño, and there is no significant impact on the deep branch.

- EP El Niño and La Niña have a clear significant impact on TCO. EP El Niño is characterized by a reduction of TCO in the tropics and an increase in middle and polar latitudes from December to March, in agreement with previous results by Cagnazzo et al., (2009) based on the Niño 3 index. The winter evolution of TCO anomalies during La Niña mirrors those found during EP El Niño in both WACCM simulations and reanalyses. In contrast, the impact of CP El Niño events on TCO is small and not significant north of the tropics. The evaluation of TCO composited anomalies for the three ENSO types in WACCM against two reanalyses and one merged satellite ozone product confirms that the model captures the main features seen in the observational datasets.

- Tropical stratospheric ozone variations are mainly driven by advection through changes in tropical upwelling that modulate the rising of ozone-poor air from the tropopause region. Our results show differences in the upwelling response between the two types of El Niño, not only in the strength but also in the timing. Changes in tropical upwelling also can lead to changes in NOx concentration, modifying the NOx ozone loss catalytic cycle, as proposed by Hood et al., (2010) and by Chipperfield et al., (1994) and Zhang et al., (2021) for QBO ozone variations. Indeed, we find a different timing in chemical anomalies in the tropical middle stratosphere (above 30 hPa), consistent with different timing in upwelling, between the two types of El Niño.

- At middle and high latitudes, mixing and advection are the main drivers of ozone variations during ENSO events in boreal winter. Regarding advection, EP El Niño events are associated with an acceleration of both shallow and deep branches of the residual circulation, which leads to an accumulation of ozone in the extratropics. In contrast, La Niña events decelerate the residual circulation and hence there is less ozone advective transport to the extratropics. Ozone advection is weak at the extratropics during CP El Niño events in agreement with the lack of impact of CP El Niño events on the deep branch of the residual circulation.

- The present study shows that the contribution of mixing processes is not negligible since its contribution to the generation of ozone anomalies at middle and high latitudes has a similar magnitude as advection. Inspection of anomalous wave dissipation patterns reveals that increased wave breaking around the polar vortex during El Niño leads to a weakening of the vortex and increased mixing across its climatological location. This leads to a decrease of ozone at the pole and an increase of ozone at mid-latitudes during boreal winter. The same mechanism, but opposite, is valid for La Niña.






We acknowledge that other sources of variability can influence the stratospheric response to ENSO and affect ozone concentrations. Most importantly, SSWs are major disruption of the polar stratosphere, and previous studies such as De La Cámara et al., (2018) and Hong and Reichler, (2021) have shown that SSWs exert a strong effect on TCO, with positive anomalies lasting more than 45 days after the SSW onset. In order to assess if the ENSO signal is different in winters with and

without SSW, we have repeated our composite analyses isolating the ENSO signal from the SSW signal. For this, we have selected the ENSO events for winters with at least one SSW occurrence and computed the composited anomalies with respect to a climatology based exclusively on winters with SSW occurrence. Analogously, we selected the ENSO events for winters without SSWs and computed the composited anomalies with respect to a climatology based on winters without SSW. This methodology is applied to each type of ENSO. The SSWs are obtained using the CP07 criterion (Charlton and Polvani, 2007).

The results obtained in both cases are similar to those shown in the analysis performed including all ENSO events (not shown). Therefore, we concluded that the ENSO signal in ozone is not significantly affected by the occurrence of SSW in our analyses. Nevertheless, we note that the relations between SSW and ENSO are complex and still under study (e.g. Song and Son, 2018). Another important source of variability in the stratosphere which could be affecting our results is the QBO (e.g. Naoe et al., (2017) , Zhang et al., (2021)). Indeed, Xie et al., (2020) showed linear interactions between the QBO and EP El Niño signals,

and its interactions in the extratropics during El Niño events were evaluated in Calvo et al., (2009). In our study, we have eliminated its influence performing a multiple linear regression analysis. However, further investigation about the joint influence of different flavors of ENSO and QBO on stratospheric ozone could be of interest since previous studies (e.g. Xie et al., 2012) have pointed out that there might be non-linear interactions between CP El Niño and QBO on the stratosphere. Finally, other aspects that might be interesting to explore would be to try to reproduce our analysis in other chemistry climate

models with a well resolved stratosphere or extend our study to the Southern Hemisphere.

*Data availability.* The output from the WACCM simulations is available at https://www2.acom.ucar.edu/gcm/ccmi-output and upon request to the corresponding author. The SWOOSH dataset is available at

https://www.esrl.noaa.gov/csd/groups/csd8/swoosh/. Data from JRA55 and MERRA2 reanalysis are freely available at https://rda.ucar.edu/datasets/ds628.1/ and at https://disc.gsfc.nasa.gov/datasets?project=MERRA-2 , respectively.

*Author contributions.* SB-B, NC and MA designed the study. SB-B performed the data analysis and wrote the article with

significant contributions from NC and MA in the interpretation of the results and reviewed the writing.



*Competing interests*. The authors declare that they have no conflict of interest.


*Acknowledgements.* SB-B acknowledges the FPU program from the Ministry of Universities (grant no. FPU19/01481). NC was supported by the Spanish Ministry of Science, Innovation and Universities through the JeDiS (RTI2018-096402-B-I00) project.

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
