# Peer review of "Driving mechanisms for the ENSO impact on stratospheric ozone"

_Atmospheric Chemistry and Physics, 2022_

## Referee Comment (RC1)

This is a very timely paper, as I think the role of ENSO diversity with respect to stratospheric dynamics (and hence transport) is a underappreciated and understudied topic. Overall I find the paper to be well-written paper with several very nice results. Most of comments are not criticisms, rather I have a series of comments that I think the authors may want to consider in terms of their interpretation of the results. The most important comment is in regards to the potential importance of PNA-like ozone teleconnections, which I think are probably quite important. To be clear, my comments are largely based on results from an ACP paper that I am the first author on, which is currently under review (see here: https://acp.copernicus.org/preprints/acp-2022-276/ ). Because this work is not yet published, I concede that the authors should not feel compelled to consider my suggestions. That said, I hope they can address at least some of the issues I raise, as I think doing so will provide readers with additional important details on the dynamics underlying ENSO-related stratospheric transport. That said, to conduct the additional analysis I suggest, the authors will need to have saved monthly mean ozone data on various pressure surfaces (geopotential height would be great too, but not required).

Also, I included some particularly relevant references at the bottom of the review that the authors probably want to glance through and then cite in the final version of their paper.

Best regards,
John Albers

**Major comments:**

Your EP-ENSO WACCM experiments are fairly similar to those we ran in the ACP paper I cited above and latitude-height cross sections seem to suggest that the stratosphere is responding similarly, at least qualitatively. For example, your Fig. 1g is more or less consistent with our Fig. 1c and 1e. There are some differences, but I am guessing that this largely represents differences in how the seasonal cycle of SSTs is prescribed in each of our respective experiments rather than actual differences in the dynamics (after all, both experiments use similar version of WACCM, so the transport dynamics should be equivalent).

Before I did into my main question about your results, I would like to make clear that I don't disagree with the part of your analysis where you diagnose the role of isentropic mixing and residual circulation advective transport. Rather, my comments below should simply be interpreted as suggestions for making your analysis and the interpretation of your results more complete. Keep in mind as you read what I write below, that I don't have a clear idea in my head for how to interpret the role of transport associated with 'ozone teleconnections' (explained below). That is, since the sum total of ozone transport should in principle be accounted for in your ozone budget equation (your Eq. 1), that would mean that the 'ozone teleconnection' related transport outlined below would (I think?) should be be accounted for by the residual advective transport terms (i.e., terms 1 and 2 on the RHS of your Eq. 1). However, I think that the residual transport terms are typically interpreted as occurring due to the induced meridional overturning (residual) circulation that is caused by wave driving. Yet in what I outline below, the ozone teleconnection-related transport

does not neatly fit into that paradigm. Indeed most review papers on stratospheric transport specifically discuss mid-stratospheric extratropical isentropic mixing and advective transport in terms of planetary scale waves #1 and 2 (e.g., Fig. 2 of Plumb 2002, https://www.jstage.jst.go.jp/article/jmsj/80/4B/80_4B_793/_article). However, in what I discuss below, the transport appears to be associated with waves with wavenumber >2, which are thus largely evanescent.

So what do I mean when I refer to an 'ozone teleconnection'? When I first looked at our own results, I assumed, as you have here, that interpreting the ENSO-forced stratospheric transport could be accomplished by diagnosing the residual circulation and eddy flux transports (i.e., your Eq. 1). However, as I started looking more closely at the ozone anomalies month-by-month on individual pressure surfaces (latitude x longitude plots), it became apparent that ENSO was forcing, largely barotropic, PNA-like ozone anomalies that extended fairly deep into the stratosphere (they are the ozone patterns that go along with the classical ENSO-forced tropical-extratropical teleconnections). At first I did not know what to think about these ozone anomalies because the BDC literature doesn't make any mention of such 'ozone teleconnections'. However, I did some digging, and it turns out that Dick Reed published a great paper back in 1950 that clearly describes this type of transport and its effect on TCO! Schoeberl and Krueger have a very nice follow-up that explains the physical processes clearly using more modern data (see both references at the bottom of this review). You can also see the signature of the teleconnections in other papers, for e.g., most importantly Zhang et al. 2015, but also Olsen et al. 2016 and Oman et al. 2013. However, none of the later three papers I just mentioned discuss the transport dynamics in terms of the ideas of Reed or Schoeberl and Krueger.

Now, as far as we could tell, the waves responsible for the teleconnections are higher wave number than #2, so they are almost certainly evanescent according to Charney/Drazin propagation criteria. Yet despite this, the waves nevertheless extend deeply into the stratosphere (at least to 20-30 hPa?). I am not sure how to refer to this physical transport mechanism, because while the waves involved may also play a role in driving various aspects of the BDC (isentropic mixing and the residual circulation), the column vertical and horizontal advection do not fit into any of the traditional BDC mechanism paradigms. Thus, in our paper we have discussed this type of teleconnection-related transport as distinct from the BDC. That said, I do not feel strongly about that interpretation, so I would leave it open to anyone else for how they want to refer to it. Indeed, I would very much like to hear a compelling argument from you and your co-authors discussing how you think this type of transport should be referred to.

So how does this apply to your paper? Well, in our paper, we are limited to one 'flavor' of ENSO. However, your data covers other ENSO flavors so it would be interesting to know how the ENSO-forced teleconnections affect transport in these other circumstances. Questions to answer and plots to consider making that would be enlightening to see might be:

- If you have the monthly data, can you plot monthly mean ozone (and geopotential height if you have it) on several pressure surfaces (say 200 hPa and 70 hPa) for a

each individual month to see how the patterns are different for the different ENSO flavors?

- Using the above plots, how does the phasing of the higher wavenumber PNA-like ozone teleconnections constructively/destructively interfere with the low-wave number climatological planetary wave pattern to produce the results you see? That is, it would seem clear that CP vs. EP El Nino and La Nina should produce quite different interference patterns and resulting transport.
- How does the seasonal cycle differ for the CP, EP, etc flavors of ENSO in terms of ozone teleconnection patterns?

Comment #1 – Figures 3 and 5: I have to admit, I find it kind of hard to envision how the individual processes unfold over the seasonal cycle using the latitudinal averages as you have done. I recognize that you are probably trying to cut down on the number of figures that you have, but I think that having figures similar to Fig. 1 and Fig. 6 make it easier to envision (spatially-temporally) how the ozone transport is unfolding. I will leave this up to the authors, but personally, I would find the it easier to understand using monthly latitude-height plots.

Comment #2 – lines: Is there much of a residual in the ozone budget equation (your Eq. 1) when you compute the individual terms? I am just wondering how well the TEM ozone budget equation closes the total ozone budget.

**Minor comments:**

Comment #1 – lines: You may want to include a reference to a paper that discusses ENSO diversity from an oceanic perspective, which would give readers better context about how diversity arises and what are its broader implications beyond the stratosphere. Personally, I think the paper by Capotondi et al. (BAMS 2015) is a good reference.

Comment #2 – line 295: I think you have a type in the text *"(not shown o complementary)."*

**References:**

Diallo, M., Konopka, P., Santee, M. L., Müller, R., Tao, M., Walker, K. A., Legras, B., Riese, M., Ern, M., and Ploeger, F.: Structural changes in the shallow and transition branch of the Brewer–Dobson circulation induced by El Niño, Atmos. Chem. Phys., 19, 425–446, https://doi.org/10.5194/acp-19-425-2019, 2019.

Rao, J., Yu, Y., Guo, D., Shi, C., Chen, D., and Hu, D.: Evaluating the Brewer–Dobson circulation and its responses to ENSO, QBO, and the solar cycle in different reanalyses, Earth and Planetary Physics, 3, 166–181, https://doi.org/10.26464/epp2019012, 2019.

Li, Y. and Lau, N.-C.: Influences of ENSO on stratospheric variability, and the descent of

stratospheric perturbations into the lower troposphere, J. Climate, 26, 4725–4748, https://doi.org/10.1175/JCLI-D-12-00581.1, 2013

Olsen, M. A., Wargan, K., and Pawson, S.: Tropospheric column ozone response to ENSO in GEOS-5 assimilation of OMI and MLS ozone data, Atmos. Chem. Phys., 16, 7091–7103, https://doi.org/10.5194/acp-16-7091-2016, 2016

Zhang, J., Tian, W., Wang, Z., Xie, F., and Wang, F.: The influence of ENSO on northern midlatitude ozone during the winter to spring transition, J. Climate, 28, 4774–4793, https://doi.org/10.1175/JCLI-D-14-00615.1, 2015

Reed, R. J.: The role of vertical motions in ozone-weather relationships, J. Atmos. Sci., 7, 263–267, https://doi.org/10.1175/1520-0469(1950)007%3C0263:TROVMI%3E2.0.CO;2, 1950.

Schoeberl, M. R. and Krueger, A. J.: Medium scale disturbances in total ozone during southern hemisphere summer, Bull. Amer. Met. Soc., 64, 1358–1365, https://doi.org/10.1175/1520-0477(1983)064%3C1358:MSDITO%3E2.0.CO;2, 1983.

Capotondi, A., Wittenberg, A. T., Newman, M., Di Lorenzo, E., Yu, J.-Y., Braconnot, P., Cole, J., Dewitte, B., Giese, B., Guilyardi, E., et al.: Understanding ENSO diversity, Bull. Amer. Met. Soc., 96, 921–938, https://doi.org/10.1175/BAMSD-13-00117.1, 2015.

Oman, L. D., Douglass, A. R., Ziemke, J. R., Rodriguez, J. M., Waugh, D. W., and Nielsen, J. E.: The ozone response to ENSO in Aura satellite measurements and a chemistry-climate simulation, J. Geophys. Res., 118, 965–976, https://doi.org/10.1029/2012JD018546, 2013.

---

## Author Comment (AC1)

We are thankful to John Albers and Peter Braesicke for their very constructive comments. We have addressed all of them and hopefully the paper is now clearer.

**REVIEWER 1 (JOHN ALBERS)**

This is a very timely paper, as I think the role of ENSO diversity with respect to stratospheric dynamics (and hence transport) is a underappreciated and understudied topic. Overall I find the paper to be well-written paper with several very nice results. Most of comments are not criticisms, rather I have a series of comments that I think the authors may want to consider in terms of their interpretation of the results. The most important comment is in regards to the potential importance of PNA-like ozone teleconnections, which I think are probably quite important. To be clear, my comments are largely based on results from an ACP paper that I am the first author on, which is currently under review (see here: https://acp.copernicus.org/preprints/acp-2022-276/ ). Because this work is not yet published, I concede that the authors should not feel compelled to consider my suggestions. That said, I hope they can address at least some of the issues I raise, as I think doing so will provide readers with additional important details on the dynamics underlying ENSO-related stratospheric transport. That said, to conduct the additional analysis I suggest, the authors will need to have saved monthly mean ozone data on various pressure surfaces (geopotential height would be great too, but not required).

Also, I included some particularly relevant references at the bottom of the review that the authors probably want to glance through and then cite in the final version of their paper.

Best regards,

John Albers

**Major comments:**

Your EP-ENSO WACCM experiments are fairly similar to those we ran in the ACP paper I cited above and latitude-height cross sections seem to suggest that the stratosphere is responding similarly, at least qualitatively. For example, your Fig. 1g is more or less consistent with our Fig. 1c and 1e. There are some differences, but I am guessing that this largely represents differences in how the seasonal cycle of SSTs is prescribed in each of our respective experiments rather than actual differences in the dynamics (after all, both experiments use similar version of WACCM, so the transport dynamics should be equivalent).

Before I did into my main question about your results, I would like to make clear that I don't disagree with the part of your analysis where you diagnose the role of isentropic mixing and residual circulation advective transport. Rather, my comments below should simply be interpreted as suggestions for making your analysis and the interpretation of your results more complete. Keep in mind as you read what I write below, that I don't have a clear idea in my head for how to interpret the role of transport associated with 'ozone teleconnections' (explained below). That is, since the sum total of ozone transport should in principle be accounted for in your ozone budget equation (your Eq. 1), that would mean that the 'ozone teleconnection' related transport outlined below would (I think?) should be be accounted for by the residual advective transport terms (i.e., terms 1

and 2 on the RHS of your Eq. 1). However, I think that the residual transport terms are typically interpreted as occurring due to the induced meridional overturning (residual) circulation that is caused by wave driving. Yet in what I outline below, the ozone teleconnection-related transport does not neatly fit into that paradigm. Indeed most review papers on stratospheric transport specifically discuss mid-stratospheric extratropical isentropic mixing and advective transport in terms of planetary scale waves #1 and 2 (e.g., Fig. 2 of Plumb 2002, https://www.jstage.jst.go.jp/article/jmsj/80/4B/80_4B_793/_article). However, in what I discuss below, the transport appears to be associated with waves with wavenumber >2, which are thus largely evanescent.

So what do I mean when I refer to an 'ozone teleconnection'? When I first looked at our own results, I assumed, as you have here, that interpreting the ENSO-forced stratospheric transport could be accomplished by diagnosing the residual circulation and eddy flux transports (i.e., your Eq. 1). However, as I started looking more closely at the ozone anomalies month-by-month on individual pressure surfaces (latitude x longitude plots), it became apparent that ENSO was forcing, largely barotropic, PNA-like ozone anomalies that extended fairly deep into the stratosphere (they are the ozone patterns that go along with the classical ENSO-forced tropical-extratropical teleconnections). At first I did not know what to think about these ozone anomalies because the BDC literature doesn't make any mention of such 'ozone teleconnections'. However, I did some digging, and it turns out that Dick Reed published a great paper back in 1950 that clearly describes this type of transport and its effect on TCO! Schoeberl and Krueger have a very nice follow-up that explains the physical processes clearly using more modern data (see both references at the bottom of this review). You can also see the signature of the teleconnections in other papers, for e.g., most importantly Zhang et al. 2015, but also Olsen et al. 2016 and Oman et al. 2013. However, none of the later three papers I just mentioned discuss the transport dynamics in terms of the ideas of Reed or Schoeberl and Krueger.

Now, as far as we could tell, the waves responsible for the teleconnections are higher wave number than #2, so they are almost certainly evanescent according to Charney/Drazin propagation criteria. Yet despite this, the waves nevertheless extend deeply into the stratosphere (at least to 20-30 hPa?). I am not sure how to refer to this physical transport mechanism, because while the waves involved may also play a role in driving various aspects of the BDC (isentropic mixing and the residual circulation), the column vertical and horizontal advection do not fit into any of the traditional BDC mechanism paradigms. Thus, in our paper we have discussed this type of teleconnection-related transport as distinct from the BDC. That said, I do not feel strongly about that interpretation, so I would leave it open to anyone else for how they want to refer to it. Indeed, I would very much like to hear a compelling argument from you and your co-authors discussing how you think this type of transport should be referred to.

So how does this apply to your paper? Well, in our paper, we are limited to one 'flavor' of ENSO. However, your data covers other ENSO flavors so it would be interesting to know how the ENSO-forced teleconnections affect transport in these other circumstances.

First of all, we would like to thank John Albers for this comment, which has stimulated an interesting discussion among the co-authors, made us re-think about our results and, in particular, about the zonally-resolved ozone transport. In addition to the response to the reviewer written below, we have included a new paragraph in the discussion section of the manuscript and a new figure (Figure 8 in the new version of the paper) to address these issues in our paper.

Questions to answer and plots to consider making that would be enlightening to see might be:

• If you have the monthly data, can you plot monthly mean ozone (and geopotential height if you have it) on several pressure surfaces (say 200 hPa and 70 hPa) for a each individual month to see how the patterns are different for the different ENSO flavors?

• Using the above plots, how does the phasing of the higher wavenumber PNA-like ozone teleconnections constructively/destructively interfere with the low-wave number climatological planetary wave pattern to produce the results you see? That is, it would seem clear that CP vs. EP El Nino and La Nina should produce quite different interference patterns and resulting transport.

• How does the seasonal cycle differ for the CP, EP, etc flavors of ENSO in terms of ozone teleconnection patterns?

First, following the reviewer's suggestions, we have plotted geopotential height and ozone anomalies at several pressure levels, in order to examine whether the ozone teleconnection patterns referred to in Albers et al., (2022) differ for ENSO flavors (we show 70 hPa and 200 hPa in Figure R1 and R2 for comparison with Figures 4 and 5 in Albers et al., (2022)). Our figures are shown for the January to March average, but we have found that the conclusions hold if we use individual months.

The anticorrelated geopotential height-ozone patterns shown in Albers et al., (2022) are consistent with those that we find both for EP and for CP El Niño events (Figure R1 and R2). However, there are some differences between EP and CP El Niño. Consistent with other results in our study, both geopotential height and ozone anomalies are weaker for CP El Niño than for EP El Niño. In fact, the anomalies within the vortex at 70 hPa are very weak for CP El Niño. On the other hand, we do not find notable differences in the seasonal evolution between EP and CP El Niño. Both ENSO flavors show an intensification of the ozone anomaly in the North Pacific throughout the winter, peaking in February-March (not shown), in agreement with Albers et al., (2022) for "canonical" El Niño.

The other question is how to interpret what the reviewer named "teleconnection-related transport". In our opinion, the ozone anomalies shown in Figures R1 and R2 represent a description of transport from a zonally resolved perspective, while the BDC transport (TEM equation) provides a zonally averaged view. As we understand it, the zonally-resolved ENSO signal on ozone is a clear fingerprint of the stationary Rossby wave train triggered by ENSO anomalies (e.g., Trenberth et al., 1998). The geopotential height

anomalies associated with these waves create a pattern of rising/sinking regions on the pressure surfaces in the upper troposphere and lower stratosphere, leading to associated "column-like" ozone anomalies, as shown in Albers et al., (2022). We agree that including this additional view can enrich our paper, and therefore we have added a new paragraph and a figure in the discussion section in the new version of the paper.

[Figure]

**Figure R1.** January-February-March composites of (left column) geopotential height and (right column) ozone mixing ratio anomalies at the 200 hPa level pressure for (top to bottom) EP El Niño, CP El Niño and La Niña.

[Figure]

**Figure R2.** As in Fig. R1 but at the 70 hPa level pressure.

On the other hand, we do not agree with the reviewer's argument that the waves being barotropic, evanescent and with wavenumbers larger than 2 imply a different type of transport not included in the TEM equation. The quasi-barotropic nature of the geopotential height anomalies is consistent with previous work (Trenberth et al., 1998 and references therein), and ozone anomalies just follow those, as argued in the previous paragraph. In addition, wavenumbers larger than 2 are expected in the lower stratosphere (e.g. Randel and Held, 1991); these waves are necessarily evanescent, as propagation conditions limit their vertical propagation, and indeed their dissipation is expected to drive the residual circulation and eddy mixing (e.g., Calvo and Garcia, 2009; Abalos et al., 2016).

In order to clarify the links between the two perspectives, we now connect the zonally-resolved ozone anomalies (teleconnection patterns in Albers et al., 2022) to the terms in the TEM continuity equation for ozone concentrations. Specifically, given that the ozone patterns can be described to first order as zonal asymmetries, we focus on the horizontal component of the TEM eddy transport/mixing term:

$$e^{\frac{z}{H}} (\cos\varphi)^{-1} (M_y \cos\varphi)_y$$

where $M_y$ is the horizontal component of the eddy transport flux:

$$M_y = -e^{\frac{-z}{H}} \left( \overline{v'O_3'} - \frac{\overline{v'T'}}{S} \overline{O_{3\,z}} \right)$$

where overbars denote zonal means, primes indicate deviations from zonal means, and subindices indicate derivatives. This term is dominated by the meridional eddy ozone flux, $\overline{v'O_3'}$.

Figure R3 shows the composites of ozone zonal anomalies, O3'. It is clear that O3' captures the wave-like nature of the O3 anomalies in Fig. R2. The black contours in Figure R3 show the corresponding composite of the eddy meridional wind anomalies, v'. The product $v'O_3'$ zonally averaged is the eddy ozone flux, and its divergence increases the zonal mean ozone concentrations in midlatitudes and decreases them at high latitudes for EP and CP El Niño; the weakening of this eddy transport during La Niña has the opposite effect (Figs. 5 to 7). What part of this eddy transport leads to irreversible mixing cannot be elucidated from this analysis.

This illustrates that the zonally-resolved ENSO signal on ozone, associated with the wave train triggered by ENSO convection, is indeed incorporated in the eddy component of BDC-transport, supporting the idea that the two (teleconnection related transport as named by Albers et al., (2022) and the transport discussed in our manuscript) represent different views of the same process. We note that Figure R3 shows this comparison in the January to March average at the 70 hPa level, but similar conclusions are reached for other or individual months and pressure surfaces.

[Figure]

**Fig. R3.** January to March composites of eddy ozone mixing ratio (colors) and eddy meridional wind (black contours) anomalies at the 70 hPa level pressure for (top to bottom) EP El Niño, CP El Niño and La Niña. Contours for eddy meridional wind are drawn every 0.5 m s-1. Solid (dashed) contours denote positive (negative) anomalies.

To complete our discussion on ozone variations during ENSO events including the zonally-resolved perspective, we have added in the discussion the following paragraph after line 471 in the new version of the manuscript:

"*While our analysis is based on the zonal mean ozone composites, studying the zonally-resolved anomalies is an interesting avenue of research, especially in the context of stratosphere-troposphere exchange. In a recent study, Albers et al., (2022) show that the zonally resolved pattern of ENSO ozone anomalies in the upper troposphere and lower stratosphere is closely connected to the geopotential height anomalies associated with the stationary Rossby wave train triggered by deep convection (e.g. Trenberth et al. 1998). In order to complement our zonal mean analysis, Figure 8 shows the zonally resolved ozone anomalies at 70 hPa, distinguishing for the first time between flavors of El Niño. Our results are highly consistent with Albers et al. (2022); they confirm the wave-like structure of the ozone anomalies and further reveal substantially larger anomalies for EP El Niño than for CP El Niño, consistent with our results. We note that the ozone zonal asymmetries evident in Fig. 8 are included in the TEM analysis used here to investigate zonal mean ENSO composites, specifically in the horizontal component of the eddy transport/mixing term in Eq. (1), given that this term is dominated by the meridional eddy ozone flux $\overline{v'O_3'}$*".

Comment #1 – Figures 3 and 5: I have to admit, I find it kind of hard to envision how the individual processes unfold over the seasonal cycle using the latitudinal averages as you have done. I recognize that you are probably trying to cut down on the number of figures that you have, but I think that having figures similar to Fig. 1 and Fig. 6 make it easier to envision (spatially-temporally) how the ozone transport is unfolding. I will leave this up to the authors, but personally, I would find the it easier to understand using monthly latitude-height plots.

In response to both reviewers, we have added Figure R4 as Supplementary Material. This figure shows latitude-pressure cross sections of the monthly mean composites from November to February of ozone mixing ratio anomalies. We do not think it is practical to show all the month-by-month figures in the paper as this would involve a very large number of plots. The total sum of the number of panels in figures 3, 4, 5 and 7 is 36 and, if we had shown this same information in monthly latitude-height plots, there would have been at least 72 panels. In addition, we believe that the latitudinal averages help to visualize some of the information, such as the downward propagation of the ozone signal at high latitudes. This is why we have preferred to keep the figures regarding mechanisms over latitudinal averages.

[Figure]

**Figure R4.** Latitude–pressure cross sections of the composites of ozone mixing ratio anomalies from November to February for (top to bottom) EP El Niño, CP El Niño and La Niña. The tropopause is indicated by the thick grey line. Black dots denote statistically significant anomalies at the 95% confidence level.

Comment #2 – lines: Is there much of a residual in the ozone budget equation (your Eq. 1) when you compute the individual terms? I am just wondering how well the TEM ozone budget equation closes the total ozone budget.

Overall, the residual term in Eq. (1) is smaller than 3% in the regions analyzed here, so it is small enough to consider that the Eq. 1 closes the total ozone budget. We have added a sentence in this regard in line 280 in the new version of the manuscript.

**Minor comments:**

Comment #1 – lines: You may want to include a reference to a paper that discusses ENSO diversity from an oceanic perspective, which would give readers better context about how diversity arises and what are its broader implications beyond the stratosphere. Personally, I think the paper by Capotondi et al. (BAMS 2015) is a good reference.

We thank the reviewer for this interesting reference. We have added it in a new paragraph after line 33 to give reader a better context about ENSO diversity.

Comment #2 – line 295: I think you have a type in the text "(not shown o complementary)."

Corrected. We meant, "Not shown".

**REVIEWER 2 (PETER BRAESICKE)**

This is a nice and interesting paper – I do not have any mayor criticism. However, I would like to share a more fundamental thought regarding the analysis method and the assumptions made and wonder if the authors could reflect on them in the discussion and or summary.

Line 38 onwards discusses the somehow contradictory nature of results to how the stratosphere behaves during El Nino's of different "flavours". However, it is very hard for the reader to understand if the classification of El Nino's is always done the same way and if composites rely on the same metrics. Personally, I like to think about El Nino's as something quite dynamic – whereas the subconscious assumption in many studies seems to imply that El Nino can be easily described in classes that lend themselves to "quasistationary composites". However, I am not surprised that different assumptions regarding the construction of such composites may lead to slightly different results (as is mentioned in line 49).

We understand the reviewer's concern and agree it is not easy to classify ENSO events into clearly different "flavors".  We refer to Capotondi et al., (2015) for a discussion on this topic including "debates on whether there are indeed two distinct modes of variability of whether ENSO can be more aptly described as a diverse continuum.". In their abstract, Capotondi and coauthors acknowledge that "ENSO events differ in amplitude, temporal evolution and spatial pattern", and recognize that a renewed interest in this diversity was stimulated after some studies (e.g. Larkin and Harrison, 2005; Ashok et al., 2007) which highlighted an unusual SST anomaly pattern associated with remote impacts (teleconnections) different from those related to typical El Niño conditions. Since then, many studies have tried to investigate the possibility of different ENSO teleconnections, including different stratospheric pathways (e.g., Hegyi and Deng, 2011; Zubiaurre and Calvo, 2012; Garfinkel et al., 2013; Hurwitz et al., 2014; Iza and Calvo, 2015; Calvo et al., 2017). In our manuscript we investigate the ENSO teleconnections further focusing on the stratospheric ozone response to different ENSO flavors, which has not received a lot of attention so far.

Figure 1 of Capotondi et al., (2015) shows the wide range of longitudes where SSTs anomalies peak during different El Niño events. They note that, despite that range of longitudes, the events at the extremes of the longitudinal distribution have distinctive characteristics which make possible to classify them into two different El Niño types. These differences do not only appear in the location and intensity of SST anomalies but also in the thermocline depth or in temporal evolution of the SST anomalies. To give the reader a better context about ENSO diversity, we have added the following paragraph after Line 33 in the new version of the manuscript:

*"Although it is widely known for many years that ENSO events are different from each other in the location and intensity of sea surface temperatures (SSTs), in recent years the importance of distinguishing between two flavors of El Niño has arisen. These two types of Niño correspond to the events in the extrema of a wide range of longitudes where SSTs anomalies peak during different El Niño events, as shown in Capotondi et al., (2015) and will be referred here as Eastern Pacific (EP) El Niño and Central Pacific (CP) El Niño. While the SSTs anomalies peak in the eastern Equatorial Pacific for EP El Niño (also*

*referred as canonical El Niño), CP El Niño (also known as El Niño Modoki or Dateline El Niño) is characterized by SSTs anomalies that peak in the central equatorial Pacific (Larkin and Harrison, 2005; Ashok et al., 2007; Kao and Yu, 2009). The differences between these two types of events do not only appear in the SSTs but also in the thermocline depth, in the development and temporal evolution of the event itself and in their remote impacts, not only in the troposphere but also in the stratosphere ((see Capotondi et al., (2020) and references therein)".*

Regarding the different methods and indices used to characterize these two types of events, and the different and sometimes confusing results published in the literature, we have rewritten lines 43 to 60 in the new version of the manuscript to make our points clearer. First of all, we would like to clarify that the controversy arises only for those events whose SSTs peak in the Central Pacific and not those in the Eastern Pacific (whose signal is robust regardless of the index and method). Second, it is true that not all the papers that we cite in Line 38 onwards in the previous version of the manuscript use the same indices to make El Niño classification. This was not a problem for EP El Niño since the signal seems to be robust but it is for CP El Niño, and it was not obvious at first (in the first papers published on this topic). Indeed, some of the works we cite were some of the first to be published on differences in stratospheric teleconnections between ENSO flavors, so it is understandable that there was no agreement regarding the classification method. In fact, the use of different methods and indices to select the CP events is one of the reasons (but not the only) why that previous works have shown different results for CP events (e.g. Garfinkel et al., 2013; Iza and Calvo, 2015). In our paper, we have decided to use the Niño3 and Niño4 indices to make the classification because in recent years most of the related works have used these indices (We have added a sentence in this regard in the method section in line 111 in the new version). Other indices commonly used to describe these two flavors of ENSO have been briefly discussed in Capotondi et al., (2015) and a sentence about this has been added as well in Line 53 in the new version of the paper.

The new paragraph (corresponding to lines 38 to 53 in the old version of the paper, 43 to 60 in the new version) is now as follows:

*"The stratospheric signal of EP El Niño is very robust and many studies have considered it as the canonical response to the warm phase of ENSO. In contrast, fewer studies have examined the NH stratospheric response to CP El Niño and their results were many times contradictory. On one hand, some studies have found a similar response to CP El Niño than that to EP El Niño in the NH polar stratosphere, that is, a weaker and warmer polar vortex (e.g., Hegyi et al., (2014), who used idealized WACCM4 simulations, or Hurwitz et al., (2014), who studied the seasonal mean polar cap geopotential anomaly at 50 hPa in a set of CMIP5 models). Other studies have also reported a weaker polar vortex during CP El Niño, but the response was significantly weaker than for EP El Niño events (Garfinkel et al., (2013); Weinberger et al., (2019)). Finally, a third group of papers have found a CP El Niño signal opposite to that of EP El Niño, albeit of smaller amplitude, (Xie et al., (2012) in reanalysis data) or not significant (Calvo et al., (2017) using a set of high-top CMIP5 models). Several reasons have been proposed to explain the contradictory results among these studies. Garfinkel et al., (2013) concluded that the sign of NH stratospheric response to CP El Niño depends on the index used to identify CP El*

*Niño events (see Capotondi et al., 2015 for a list of the main indices used in the literature), the composite size and the month average analyzed. Note that, since the studies cited above do not use the same methodology or the same indices to classify ENSO events into EP or CP El Niño, it is not surprising that differences appear between their results in the response to CP El Niño. Calvo et al., (2017) highlighted the importance of studying the seasonal evolution of the NH stratospheric signals for understanding the different EP and CP El Niño impacts. Other reasons may include interactions between El Niño and the Quasi-Biennial Oscillation (QBO; Xie et al., 2012) and overlapping with the signal from Sudden Stratospheric Warmings (SSWs; Iza and Calvo, 2015). Overall, further investigation is still needed to better understand the differences between EP and CP El Niño signals on the NH stratosphere".*

The choice of NDJF as a season is logical, if one subscribes to a quite stationary picture of the "ENSO" classes. Here, I would find it useful to reflect a little more on the choice of this very long (and slightly artificial) season. True, averaging might make a problem more linear, but could I get away with less averaging? In other words: Is there a dominant feature in the NDJF seasons that determines the long term mean? The question seems an interesting one to me, because you are presenting time-dependent composites (from October to May) that show clear sign changes in NDJF (or, I am not getting the colour scale right). Here I would really like to see some explanation how the extended season and the time-dependent composites relate to each other.

It is true that by computing "seasonal" averages (in our case NDJF) we lose some information on the differences between the EP and CP El Niño impacts. This has been pointed out before, see for instance Calvo et al., (2017) who highlights the importance of studying the seasonal evolution to understand the differences between EP and CP El Niño events at high latitudes. Consistently, we agreed to show the temporal evolution when possible in our paper. Indeed, this is done in all the figures of the manuscript except in Figures 1 and 6. These two figures complement the time-dependent figures by including the latitudinally-resolved anomalies. In order to find a reasonable number of figures and multiple panels in the manuscript, we decided to compute seasonal averages in Figures 1 and 6. However, as it may be helpful for the reader to see the month by month figures for ozone anomalies as a complement to Figures 1 and 3, we have added them in Figure S1 in the Supplementary Material in our new version of the paper. The new figure is also included here above in response to the other reviewer (Figure R4).

The main purpose of Figure 1 is to compare our ENSO signal in temperature (T), zonal wind (U), residual circulation (through w*) and ozone with results from previous literature. Indeed, the ENSO signal on T, U and w* have been widely investigated. The results of this comparison (see lines 154 to 205 in the new version of the manuscript) give us confidence that the WACCM4 simulations used in this work correctly reproduce the behavior of the stratosphere during ENSO events before analyzing the ENSO impact on ozone and the underlying mechanisms in more detail. We believe the use of the NDJF average is valid for this purpose, as we see that the main well-known characteristics are well reproduced in these simulations (e.g., a warmer polar stratosphere and a weaker polar vortex in the case of "canonical" El Niño (Calvo et al., 2010; Mezzina et al., 2021)).

Our extended season includes November because we find significant ozone anomalies in extratropical latitudes already in this month as can be seen below in Figure R4. We added this sentence in Line 157 to clarify our choice: "*We have included November in the extended winter season because we found significant ozone anomalies in extratropical latitudes already in this month (Fig. S1)*".

The comparison of Figure R4 with Figure 1g-i illustrates the relationship between the extended season and the time-dependent composites in the case of ozone. In the tropics and mid-latitudes, the seasonal mean and the temporal evolution give almost the same information. While the exact location and extent of the anomalies change over time, the patterns are consistent and there is no change in sign. Not surprisingly, it is at high latitudes where the month to month differences are the largest. The ozone signal propagates downwards from the middle stratosphere at the beginning of winter to the lower stratosphere at the end of winter. Note that this downward propagation at high latitudes is already clearly seen in Figure 3g-i in the manuscript. When computing the seasonal mean, we are averaging anomalies of opposite sign in certain regions, smearing out the signal. However, the key characteristics can still be appreciated in the seasonal mean composited anomalies in Fig. 1: more ozone at the pole during EP El Niño than CP El Niño, and less ozone during La Niña.

We believe Fig. 1 still gives a good overview of the ozone and circulation anomalies associated with the ENSO flavors, while we have added Figure S1 (R4 here) to show the monthly mean evolution of the signal to complement Figure 3 which already shows monthly mean evolutions at certain latitudinal averages. The relevant time-dependence information which can be therefore extracted from Figures like R4 is already included in the manuscript in Figures 3, 4, 5 and 7.

When talking about statistical significance it would be helpful to state what kind of variability is forming the baseline – I am not always 100% percent sure if you base all assessments on the full ensemble information (or on individual realisations) – or just the case-by-case standard variability of the different ensemble members. Please clarify where appropriate.

We thank the reviewer for pointing this out. We agree this was missing in the submitted version. For all the calculations in the manuscript (including significance) we used information from all ensemble members and do not work with the ensemble mean. This means that, for each simulation we calculate the anomalies with respect to the climatology of that simulation (as explained in line 121 in the new version of the manuscript), then we identified the ENSO events in this simulation and finally composited all ENSO events in the four simulations we are analyzing.  For the Monte Carlo test, we applied the same procedure. As the composite is made with the information of the four ensemble members, the random selection during the Monte Carlo test is made from the 240 years of the four ensemble members (60 years for 4 simulations). Thus, we consider together the anomalies of all ensemble members and randomly select as many years from the total of 240 years as there are cases in the ENSO composite of anomalies (which will depend on each ENSO type). Then, we composite that random selection and repeat the process 1000 times to create a "composites distribution". The anomaly is considered significant when it is outside of the central 95% of this distribution.

We have modified the text between lines 120 and 130 in the new version (lines 110 to 115 in the old version) of the manuscript to clarify the methodology:

"*The ENSO signal is analyzed by compositing monthly mean anomalies for the identified ENSO events (Table 1) in boreal extended winter (October to March). For each ensemble member, anomalies are computed with respect to a 21-year running mean climatology of that member, which allows to remove possible linear and non-linear trends. This is particularly important in the case of ozone since Ozone Depleting Substances (ODSs) concentrations are not uniform throughout the 1955-2014 period. After that, the ENSO events are identified in each simulation and finally composited all ENSO events from the four simulations analyzed. The statistical significance of the ENSO signal in the composites is assessed with a Monte Carlo test of 1000 trials at the 95 % confidence level. To do so, we consider together the anomalies of all ensemble members and randomly select as many years from the total of 240 years (60 years for four simulations) as there are cases in the composite of anomalies (which depends on each type of ENSO). We composite this random selection and repeat the process 1000 times to create a composites distribution. The anomaly is considered significant when it is outside of the central 95% of the random distribution.*"

In addition, some small thoughts regarding the ozone impact and a more continuous approach to the (stratospheric) ENSO impact. Because the thoughts are documented by some "old" papers I (co-)wrote, I have submitted the review under my name:

Thanks for these interesting references. We included the paper by Pyle, Braesicke and Zeng (2005) in the Introduction, when discussing previous works which showed the ENSO impact on ozone in the tropics (Line 66 in the new version of the paper).

1) A strong anticorrelation between vortex strength and (total)ozone is somehow an intrinsic feature (regardless of ENSO state, e.g., figure 1 in doi:10.1029/2002GL015973). However, ENSO can certainly impact part of the signal when either the vortex strength and/or the equatorward transport of "ozone rich air" is modified. Even though such relationship can be established with a simple stratospheric wind metric, the underlying contribution for the ozone change is largely coming from a region that is suggestive of the shallow branch of the BDC (e.g., figure 3 (right panel in doi:10.1029/2002GL015973)). This illustrates the importance of monthly extremes as dominant features in an extended season. Thus, my point above regarding the extended season – is there a dominant effect / month in the extended season?

As shown in new Figure S1-R4 and Figure 3 of the manuscript, we have not found a particular month which dominates the ENSO signal on ozone. ENSO signal in ozone appears throughout the entire extended winter season with a seasonal evolution. When we analyze the operating mechanisms, which directly assess the impact on ozone of changes in mixing across the polar vortex and in the residual circulation (Figures 4, 5 and 7), we also found that there is no particular month that dominates over the rest of the months. The anomalies peak in different months depending on the region and the type of ENSO.

2) To avoid a strong quasi-stationary assumption a more continuous approach to diagnosing ENSO impacts could be useful that could involve lagged correlations. For example, Figure 4 in doi:10.1039/b417947c illustrates such a simple approach, using lagged correlations between the NINO3 index and three different latitudinal regions. It is presumably not surprising that the correlation magnitude is largest in the tropics, and that the relative extrema in the mid-latitudes lag by a couple of months. However, this behaviour implies to me that maybe composites might be a little flawed. However, I do admit that the mentioned example also requires more work to provide conclusive evidence of how a more continuous approach might help to unravel different ENSO classes.

We understand the reviewer's concern. However, the use of correlations would limit us to analyze only the linear interactions between ENSO and stratospheric ozone. In contrast, by analyzing composites, we are also including possible non-linearities. Indeed, by focusing on the temporal evolution of the anomalies in different regions (Figures 2,3,4,5,7) we can already infer the time-dependent characteristics in the different regions. For example, in the tropical region, ozone anomalies appear already in October (Figures 2 and 3), which agrees with a positive correlation with SST anomalies at zero lag (or near to zero), since SST anomalies typically peak in early winter. On the other hand, both in mid-latitudes and at the pole the ozone anomalies appear from November onwards, and typically peak at the end of the winter (Figures 2 and 3), which implies a lag of a few months. These results are consistent with the findings in Pyle et al., (2005) using lagged correlations.

Just to be clear: I am not expecting the papers to be cited – I would just like to trigger some (more) critical reflection regarding the composite method – either in the introduction or the summary. Otherwise, I am very happy to accept the findings of the study as they are (it all comes back to the "contradictory" statement in line 39 – in conjunction with a very extended season and "case" composites that show some significant changes in the extended season).

Because this review is more a request for additional information / clarification I abstain from minor comments and hope that the authors will deliver a small update to this nice paper in due course.

**REFERENCES**

Abalos, M., Legras, B., and Shuckburgh, E.: Interannual variability in effective diffusivity in the upper troposphere/lower stratosphere from reanalysis data, Q. J. R. Meteorol. Soc., 142, 1847–1861, https://doi.org/10.1002/qj.2779, 2016.

Albers, J. R., Butler, A. H., Langford, A. O., Elsbury, D., and Breeden, M. L.: Dynamics of ENSO-driven stratosphere-to-troposphere transport of ozone over North America, Atmos. Chem. Phys., 22, 13035–13048, https://doi.org/10.5194/acp-22-13035-2022, 2022.

Ashok, K., Behera, S. K., Rao, S. A., Weng, H., and Yamagata, T.: El Niño Modoki and its possible teleconnection, J. Geophys. Res. Ocean., 112, 1–27, https://doi.org/10.1029/2006JC003798, 2007.

Calvo, N. and Garcia, R. R.: Wave forcing of the tropical upwelling in the lower stratosphere under increasing concentrations of greenhouse gases, J. Atmos. Sci., 66, 3184–3196, https://doi.org/10.1175/2009JAS3085.1, 2009.

Calvo, N., Garcia, R. R., Randel, W. J., and Marsh, D. R.: Dynamical mechanism for the increase in tropical upwelling in the lowermost tropical stratosphere during warm ENSO events, J. Atmos. Sci., 67, 2331–2340, https://doi.org/10.1175/2010JAS3433.1, 2010.

Calvo, N., Iza, M., Hurwitz, M. M., Manzini, E., Peña-Ortiz, C., Butler, A. H., Cagnazzo, C., Ineson, S., and Garfinkel, C. I.: Northern hemisphere stratospheric pathway of different El Niño flavors in stratosphere-resolving CMIP5 models, J. Clim., https://doi.org/10.1175/JCLI-D-16-0132.1, 2017.

Capotondi, A., Wittenberg, A. T., Newman, M., Di Lorenzo, E., Yu, J. Y., Braconnot, P., Cole, J., Dewitte, B., Giese, B., Guilyardi, E., Jin, F. F., Karnauskas, K., Kirtman, B., Lee, T., Schneider, N., Xue, Y., and Yeh, S. W.: Understanding enso diversity, Bull. Am. Meteorol. Soc., 96, 921–938, https://doi.org/10.1175/BAMS-D-13-00117.1, 2015.

Capotondi, A., Wittenberg, A. T., Kug, J. S., Takahashi, K., and McPhaden, M. J.: ENSO Diversity, Geophys. Monogr. Ser., 253, 65–86, https://doi.org/10.1002/9781119548164.ch4, 2020.

Garfinkel, C. I., Hurwitz, M. M., Waugh, D. W., and Butler, A. H.: Are the teleconnections of Central Pacific and Eastern Pacific El Niño distinct in boreal wintertime?, Clim. Dyn., 41, 1835–1852, https://doi.org/10.1007/s00382-012-1570-2, 2013.

Hegyi, B. M. and Deng, Y.: A dynamical fingerprint of tropical Pacific sea surface temperatures on the decadal-scale variability of cool-season Arctic precipitation, J. Geophys. Res. Atmos., 116, 1–13, https://doi.org/10.1029/2011JD016001, 2011.

Hegyi, B. M., Deng, Y., Black, R. X., and Zhou, R.: Initial transient response of the winter polar stratospheric vortex to idealized equatorial pacific sea surface temperature anomalies in thwe NCAR WACCM, J. Clim., 27, 2699–2713, https://doi.org/10.1175/JCLI-D-13-00289.1, 2014.

Hurwitz, M. M., Calvo, N., Garfinkel, C. I., Butler, A. H., Ineson, S., Cagnazzo, C., Manzini, E., and Peña-Ortiz, C.: Extra-tropical atmospheric response to ENSO in the CMIP5 models, Clim. Dyn., 43, 3367–3376, https://doi.org/10.1007/s00382-014-2110-z, 2014.

Iza, M. and Calvo, N.: Role of Stratospheric Sudden Warmings on the response to Central Pacific El Niño, Geophys. Res. Lett., https://doi.org/10.1002/2014GL062935, 2015.

Kao, H. Y. and Yu, J. Y.: Contrasting Eastern-Pacific and Central-Pacific types of ENSO, J. Clim., 22, 615–632, https://doi.org/10.1175/2008JCLI2309.1, 2009.

Larkin, N. K. and Harrison, D. E.: Global seasonal temperature and precipitation anomalies during El Niño autumn and winter, Geophys. Res. Lett., 32, 1–4, https://doi.org/10.1029/2005GL022860, 2005.

Mezzina, B., Palmeiro, F. M., García-Serrano, J., Bladé, I., Batté, L., and Benassi, M.: Multi-model assessment of the late-winter stratospheric response to El Niño and La Niña, Clim. Dyn., https://doi.org/10.1007/s00382-021-05836-3, 2021.

Pyle, J. A., Braesicke, P., and Zeng, G.: Dynamical variability in the modelling of chemistry-climate interactions, Faraday Discuss., 130, 27–39, https://doi.org/10.1039/b417947c, 2005.

Randel, W. J. and Held, I. M.: Phase speed spectra of transient eddy fluxes and critical layer absorption, https://doi.org/10.1175/1520-0469(1991)048<0688:PSSOTE>2.0.CO;2, 1991.

Trenberth, K. E., Branstator, G. W., Karoly, D., Kumar, A., Lau, N. C., and Ropelewski, C.: Progress during TOGA in understanding and modeling global teleconnections associated with tropical sea surface temperatures, J. Geophys. Res. Ocean., 103, 14291–14324, https://doi.org/10.1029/97jc01444, 1998.

Weinberger, I., Garfinkel, C. I., White, I. P., and Oman, L. D.: The salience of nonlinearities in the boreal winter response to ENSO: Arctic stratosphere and Europe, Clim. Dyn., 53, 4591–4610, https://doi.org/10.1007/s00382-019-04805-1, 2019.

Xie, F., Li, J., Tian, W., Feng, J., and Huo, Y.: Signals of El Niño Modoki in the tropical tropopause layer and stratosphere, Atmos. Chem. Phys., 12, 5259–5273, https://doi.org/10.5194/acp-12-5259-2012, 2012.

Zubiaurre, I. and Calvo, N.: The El Niño-Southern Oscillation (ENSO) Modoki signal in the stratosphere, J. Geophys. Res. Atmos., 117, 1–15, https://doi.org/10.1029/2011JD016690, 2012.